# Integrating multiple sources of ecological data to unveil macroscale species abundance

Keiichi Fukaya [1,2 ✉], Buntarou Kusumoto [3], Takayuki Shiono [3], Junichi Fujinuma [3] & Yasuhiro Kubota [3]

The pattern of species abundance, represented by the number of individuals per species within an ecological community, is one of the fundamental characteristics of biodiversity. However, despite their obvious significance in ecology and biogeography, there is still no clear understanding of these patterns at large spatial scales. Here, we develop a hierarchical modelling approach to estimate macroscale patterns of species abundance. Using this approach, estimates of absolute abundance of 1248 woody plant species at a 10-km-grid-square resolution over East Asian islands across subtropical to temperate biomes are obtained. We provide two examples of the basic and applied use of the estimated species abundance for (1) inference of macroevolutionary processes underpinning regional biodiversity patterns and (2) quantitative community-wide assessment of a national red list. These results highlight the potential of the elucidation of macroscale species abundance that has thus far been an inaccessible but critical property of biodiversity.

[1] National Institute for Environmental Studies, 16-2 Onogawa, Tsukuba, Ibaraki 305-8506, Japan. [2] The Institute of Statistical Mathematics, 10-3 Midoricho, Tachikawa, Tokyo 190-8562, Japan. [3] Faculty of Science, University of the Ryukyus, 1 Senbaru, Nishihara, Okinawa 903-0213, Japan. ✉email: fukaya.keiichi@nies.go.jp

A better understanding of global patterns of species commonness and rarity has been a fundamental requirement in ecology and evolutionary biology since the time of Darwin[1–4]. Nonetheless, we still lack a clear understanding of the patterns of species abundance, especially at large spatial scales. Species abundance observed at broad spatial scales is itself of ecological relevance as a fundamental property of biodiversity but also can largely contribute to gaining a deeper understanding of the structure and dynamics of local communities. As one example, the concept of a species pool and metacommunity, as a source of species in the process of local community assembly, is considered to reflect species diversity at a broad spatial scale[5–7]. Nonetheless, the properties of ecological communities are difficult to measure directly at large spatial scales, thereby hindering analyses of the patterns of species diversity[8]. According to ecological theories, the species abundance distributions (SADs), measured as the relative or absolute number of individuals per species within an ecological community[9,10], at broad spatial scales may even provide clues into the evolutionary underpinnings of biodiversity; the unified neutral theory of biodiversity and biogeography (UNTB) predicts that the SAD in a metacommunity is dependent on the dominant mode of speciation occurring in the metacommunity[5,11,12]. This implies that the macroevolutionary driver of regional biodiversity may be inferred by analysing the SADs across an appropriate spatial scale that represents a metacommunity. Moreover, species abundance determined at large spatial scales may have even greater significance for the conservation of biodiversity, in which a global or regional assessment of species rarity is a fundamental step for evaluating extinction risk (e.g. compilation of a red list[13]) and in determining conservation priority. Despite this well-recognised considerable importance of species abundance at large spatial scales, current empirical knowledge on species abundance is limited to local scales, which is likely due to the extensive survey effort required for counting every diverse species in a community.

In this view, we develop a hierarchical modelling framework[14–16] that estimates species abundance over a large geographic extent, which we named "macroscale species abundance". Given the expense associated with the collection of individual count data at the community level, we adopt a modelling approach that utilises spatially replicated multispecies detection–nondetection observations and information on the geographical distribution of species, which can be potentially obtained with less survey effort over a large geographic breadth. On the basis of these data, the proposed model enables estimating the individual density of each species within each defined geographical unit; the resulting estimates can then be used to obtain an abundance map of each species along with maps indicating the properties of the community, such as species diversity and community size, providing valuable information for understanding the facets of regional biodiversity and for determining conservation priorities (Fig. 1). We apply the model to a large dataset of woody plant communities in mid-latitude forests on East Asian islands, including the Japanese archipelago. The dataset comprises 40,516 vegetation survey records and geographical ranges of species from various data sources, which were used to estimate the abundance for 1248 species within 4619 10-km grid squares. We then used the estimated species abundance for two post-hoc analyses, including an inference on the macroevolutionary processes of regional biodiversity and an assessment of the completeness of the national red list for covering potentially threatened species. In the former, analysis of the SADs determined at large spatial scales based on the UNTB provide strong support for protracted speciation as the dominant mode of speciation in the metacommunity. Furthermore, this approach yield an estimate of the speciation rate in four insular ecoregions, highlighting the importance of geographic characteristics in macroevolutionary

processes, and predicted the average species lifetime that was congruent with previous estimates. In the latter analysis, the estimated abundance and area of occupancy of each species illuminate an important biodiversity conservation gap that has been potentially overlooked: i.e. some species with limited abundance and area of occupancy may have been underrated in a national red list. These examples highlight the potential of our proposed approach for the less expensive elucidation of macroscale species abundance.

## Results

**Species abundance of woody plants in East Asian islands.** The result highlighted geographical and latitudinal patterns of biodiversity over the East Asian islands (Fig. 2). The total abundance of woody plants (which encompasses introduced species and nominally includes individuals of any size; see Woody plant communities in East Asian islands in the "Methods" section) revealed no apparent distinct latitudinal patterns, although it tended to be slightly smaller at lower latitudes where few large islands exist (Fig. 2a). By contrast, species richness and diversity index (represented by Shannon entropy) exhibited a clear, and similar, hump-shaped latitudinal gradient: species diversity was highest in the mid-latitude zone of the Japanese archipelago, which has a substantial amount of land area, and decreased in both north and south directions (Fig. 2b, c). We observed that compared to species richness, the diversity index shows a less-autocorrelated geographical pattern (Fig. 2c).

Variation in the individual density of species, conditional on presence, and occurrence probability of species were explained by two cell-specific covariates, actual evapotranspiration (AET), and the human influence index (HII), along with their interaction effect and correlated random cell effects. Conditional density decreased with AET whereas it increased with HII, except under relatively high AET value (Supplementary Fig. 3a). Occurrence probability increased with AET, whereas it decreased with HII (Supplementary Fig. 3b). Random cell effects of conditional density and occurrence probability were negatively correlated (Supplementary Note 2).

The estimates of species abundance were validated based on independent local abundance datasets of woody plant communities obtained in forest inventory plots (FDP, NFI, and FSLE). This validation has confirmed a positive correlation between the predicted and observed log density of woody plant species (Fig. 3 and Table 1). The model prediction of individual density was biased positively for FDP and NFI but negatively for FSLE, which also indicated a relatively greater root mean square error (Table 1). A species-base assessment of the predictive performance showed that a relatively smaller number of species with higher individual density and/or narrower geographic range had a particularly large negative bias (Supplementary Fig. 4).

The total density of woody plants was validated at the scale of the grid cells based on an independent global estimate of tree density (the global map of forest trees, GMFT[17]). This validation indicated a strong correlation of the log individual density per 10-km grid cell between the fitted model and GMFT, although the correlation was rather unnoticeable when examined with log individual density per unit area of natural forest (Fig. 3 and Table 1). The prediction of the fitted model to the GMFT estimates was positively biased (Table 1). As a result, the fitted model estimated the total woody plant abundance within the natural forest in Japan to ~21.0 (standard error: 1.5) billion ($10^9$), which was about 56% higher than the GMFT estimate in the same area (13.5 billion).

The abundance of individual species ranged over six orders of magnitude, from species with $10^8$ individuals to species with hundreds of individuals. Maps of the estimated abundance

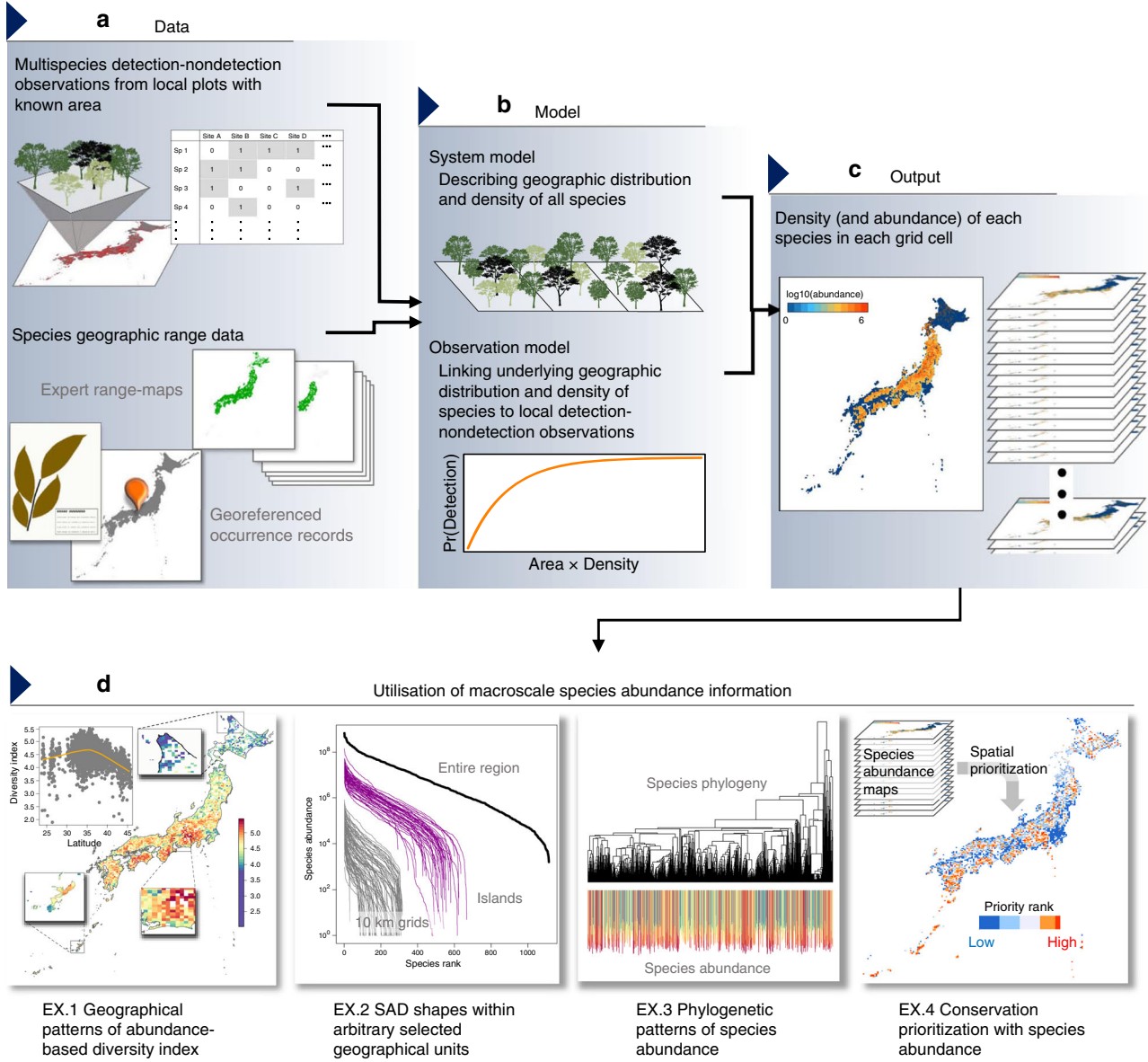

**Fig. 1 A framework for estimation of macroscale species abundance.** Spatially replicated detection–nondetection observations and various information on species geographic distribution **a** are integrated in a hierarchical model that links binary observations to underlying species abundance **b**. A model fitting yields estimates of individual density of each species in each geographic grid cell, which can then be used to derive estimates of species abundance with the area of suitable habitat **c**. The results can be used for diverse purposes relevant to, e.g. community ecology, macroecology, biogeography, and applied fields of ecology **d**.

of each species are provided in Supplementary Note 4. In the following subsections, we describe two post hoc analyses that highlight the utility of macroscale species abundance estimates.

**Inferring macroevolutionary processes of biodiversity.** The UNTB predicts that the statistical form of the SAD in the metacommunity is intimately linked to the mode of speciation[5,11,12,18–20]. This theoretical foundation enables utilizing the UNTB to infer the role of macroevolution in shaping ecological patterns based on statistical analyses of SADs. Nevertheless, the fact that species abundance data can be obtained only from local communities has been a critical limitation to the practical inference of macroevolutionary processes[5,11,12,18]. Preparing estimates of macroscale species abundance may prove to be a useful and probably only solution to this problem, given that

obtaining data on species abundance over a huge spatial extent is obviously unrealistic.

We obtained metacommunity SADs for four ecoregions that belong to different biogeographic groups (i.e. the central continental arc, northern continental arc, southern continental arc, and oceanic islands; Fig. 4) by aggregating abundance estimates over grid cells within each region. For each ecoregion, three variants of the UNTB, the point mutation speciation model[5,21], random fission speciation model[12], and protracted speciation model[11], were fitted to make an inference about the metacommunity SAD. Additional details of this analysis are described in the "Methods" section (subsection, Inference of macroevolutionary processes in metacommunities).

The SADs of metacommunities in the four ecoregions followed a left-skewed, lognormal-like distribution, whose short left tail indicates that the number of very rare species was negligible (Fig. 4). Among the three variants of the UNTB, this pattern of

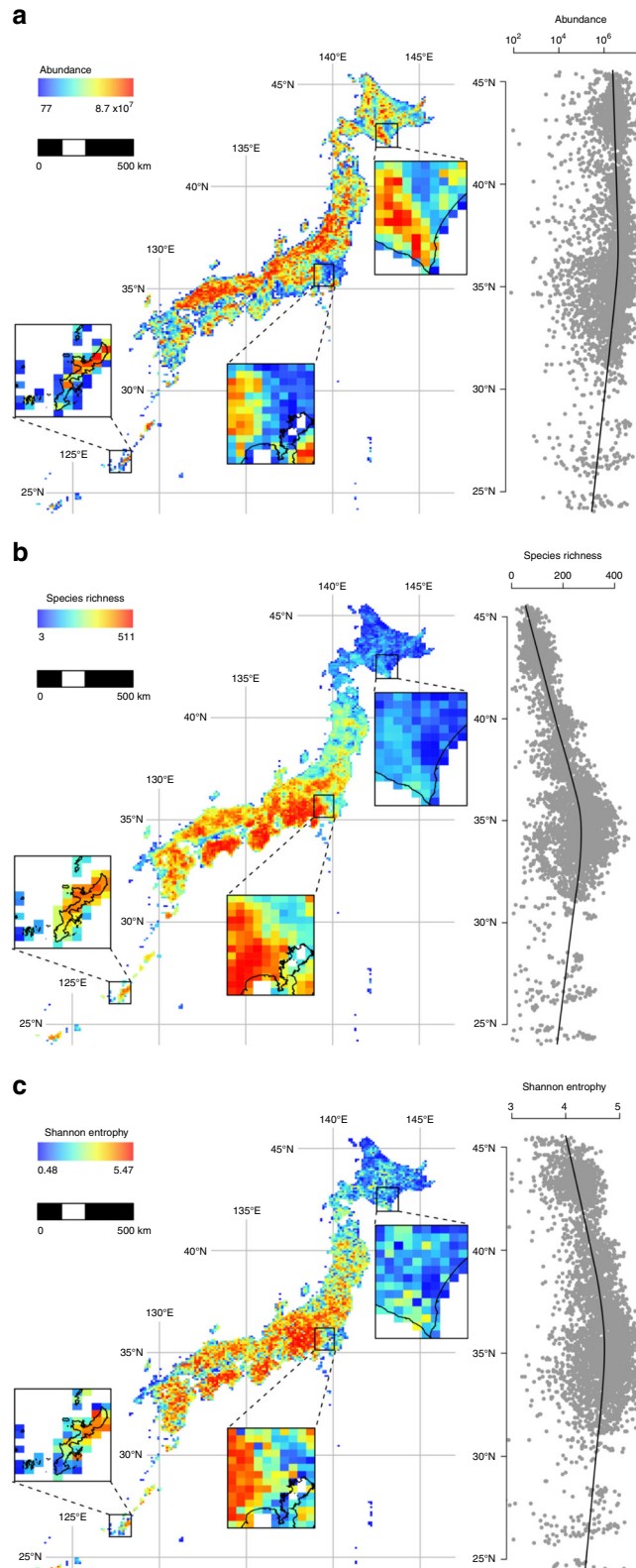

**Fig. 2 Maps of community properties estimated in 10-km grid cells. a** Total number of individuals (abundance), **b** number of species (species richness) and **c** species diversity index (Shannon entropy). To illustrate finer spatial patterns, three arbitrarily selected sections are enlarged. The marginal plots indicate estimated values in each grid cell along latitude, with a fitted line obtained using a lowess smoother. A scatter plot matrix for these three variables is provided in Supplementary Fig. 6. Maps of standard errors of these three variables are provided in Supplementary Fig. 7.

the metacommunity SADs was consistently well explained by the protracted speciation model (Table 2 and Fig. 4). Point mutation speciation model fitted well at the largest abundance classes, but failed to predict the number of less common species and rare species. Random fission speciation model overpredicted the number of moderately abundant species, while underpredicting the number of less common species. The results suggest that the manner of species diversification in these metacommunities was represented by neither of the two extreme modes, point mutation speciation or random fission speciation, but by an intermediate process expressed as a protracted speciation.

The macroscale species abundance yielded estimates of the metacommunity size $J_M$ for each ecoregion, which enabled us to disentangle speciation rate $\nu$ from the fundamental biodiversity number $\theta$ (Table 3). A higher speciation rate and shorter average lifetime of a species ($L$) was observed in ecoregions composed of small and isolated islands, the oceanic islands region, and the southern continental arc region (Table 3), implying relatively rapid evolutionary turnover of the metacommunity in those regions. The magnitude of $L$ largely differed between the models; the point mutation speciation model predicted an average species lifetime of <20 generations, while the random fission speciation model predicted a very long lifetime, up to tens of millions of generations. Assuming that the average generation time of woody plants is about 30 years[22,23], the estimates of lifetime (i.e. hundreds of years in the point mutation speciation model and up to hundreds of millions of years in the random fission speciation model) are ecologically unrealistic. In contrast, the protracted speciation model provided moderate estimates of $L$ that range from hundreds of thousands of years to tens of millions of years, which are comparatively congruent with previous estimates for species lifetime of vascular land plants based on fossil records[24,25].

**Implications for biodiversity conservation.** Red listing of threatened species is a process of prioritisation that is a fundamental step of conservation planning. Modern red lists, including the IUCN Red List[13] and a number of national red lists[26], are based on quantitative assessments of rates of population decline, extinction risk, area of species range, and population abundance. Nonetheless, such assessments are difficult to conduct for every species; as a result, red lists are largely incomplete, with several potentially threatened species left unevaluated[27]. Evaluation of macroscale species abundance is likely to mitigate this limitation in the process of red listing of species by facilitating the assessment of spatial distribution of abundance for a broad range of species.

We summarised the abundance and area of occupancy of species with respect to their category in the national red list of vascular plants in Japan, which essentially adheres to the IUCN Red List criteria but comprises modified categories[28]. Details of the categories in this list are described in the "Methods" section (subsection, National red list categories of species in Japan). We obtained the regional abundance of each species by aggregating the abundance estimates over all grid cells. In addition, for each species, we obtained the area of occupancy within the region by summing the area of natural forest over all grid cells weighted by the estimated posterior probability of their presence.

We observe that the categories of the national plant red list adequately reflect the magnitude of abundance and area of occupancy of species in the country (Fig. 5). Nevertheless, the result indicated that the NC category (Not Classified; species that are not classified to the at-risk categories) contains a number of species with a limited number of individuals and area of occupancy. Most of these species may not be classified in

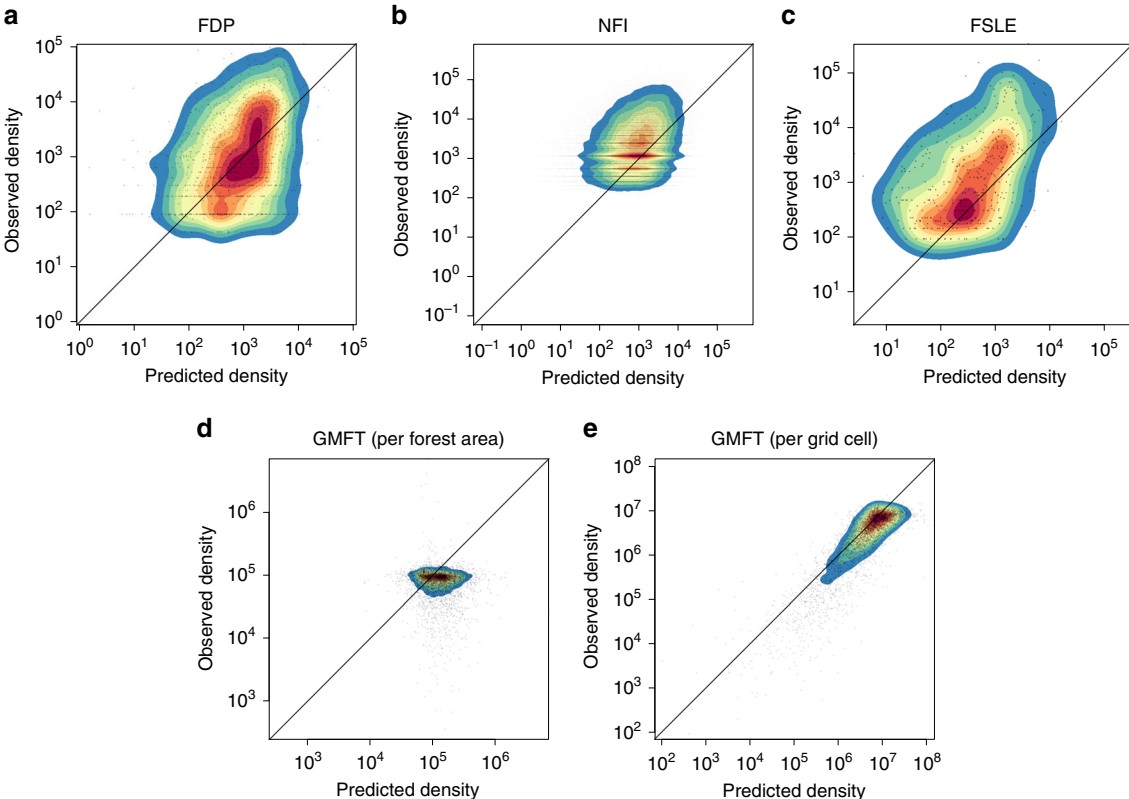

**Fig. 3 Model prediction of individual density to validation data.** **a** FDP, **b** NFI, **c** FSLE, **d** GMFT (per forest area) and **e** GMFT (per grid cell). Scatter diagrams, along with kernel density estimate, of the logarithms of individual density per 1 km² natural forest between the model prediction and the four validation datasets (FDP, NFI, FSLE, and GMFT). Reddish colours indicate higher density. The diagonal lines are the identity lines. For GMFT, in which individual density was validated at the scale of the 10-km grid cell, a scatter diagram based on individual density per grid cell is also shown in panel **e**. Coefficients of correlation are given in Table 1. The relationships are also shown on the arithmetic scale in Supplementary Fig. 8. Abbreviations: FDP – forest dynamics plots, NFI – National forest inventory, FSLE – forest sampling plots along latitudinal and elevational gradients, GMFT – global map of forest trees.

| Table 1 Predictive performance of the fitted model. | | | |
| --- | --- | --- | --- |
| **Dataset** | **RMSE** | **Bias** | **Corr** |
| FDP | $2.85 \times 10^3$ | $5.48 \times 10^1$ | 0.358 |
| | $[0.32 \times 10^3]$ | $[3.82 \times 10^1]$ | $[0.037]$ |
| NFI | $2.85 \times 10^3$ | $2.72 \times 10^1$ | 0.276 |
| | $[0.18 \times 10^3]$ | $[2.48 \times 10^1]$ | $[0.034]$ |
| FSLE | $5.84 \times 10^3$ | $-3.44 \times 10^2$ | 0.485 |
| | $[0.25 \times 10^3]$ | $[0.66 \times 10^2]$ | $[0.053]$ |
| GMFT | $1.62 \times 10^5$ | $5.08 \times 10^4$ | $-0.063$ |
| | $[0.05 \times 10^5]$ | $[0.90 \times 10^4]$ | $[0.019]$ |
| | | | 0.868 |
| | | | $[0.004]$ |

The performance of the fitted model to predict individual density per 1 km² natural forest was examined with four validation datasets (FDP, NFI, FSLE, and GMFT) and three benchmarks (RMSE, Bias, and Corr). The individual density of species was validated with FDP, NFI, and FSLE dataset, whereas the total density of woody plants was validated with GMFT dataset. For GMFT, in which individual density was validated at the scale of the 10-km grid cell, the correlation coefficient calculated based on individual density per grid cell is also shown in the second row. Standard errors are shown in brackets.
Abbreviations: FDP – forest dynamics plots, NFI – National forest inventory, FSLE – forest sampling plots along latitudinal and elevational gradients, GMFT – global map of forest trees, RMSE – root mean square error of the model prediction for individual density, Bias – bias of the model prediction for individual density, Corr – correlation coefficient of the log individual density between model prediction and validation dataset.

threatened categories immediately because the IUCN Red List criteria based on range size and population size require additional conditions, such as population decline and habitat fragmentation; species with <20 km² area of occupancy can be classified,

however, as Vulnerable (VU) according to the IUCN Red List D2 criterion[13]. The result thus implies that there are a number of species for which potential threats of extinction may have been overlooked.

## Discussion

We developed a modelling approach to estimate macroscale species abundance, which has thus far been an inaccessible property of biodiversity. These estimates are critically informative in both basic and applied fields of ecology and biogeography, as demonstrated through the application of this approach with data on woody plants in East Asian islands. The approach relies on multispecies detection–nondetection observations collected by means of planned surveys to estimate macroscale species abundance. Moreover, various forms of auxiliary data that indicate the geographic distribution of species can be integrated to explicitly account for the range of species, which can even result in an improvement of the inference (see Supplementary Note 2). As compared to community-level individual count data such as forest inventory, these types of data can be obtained in greater amounts with less effort and/or can be collected over a larger geographic extent. Therefore, the proposed approach can facilitate our understanding of biodiversity patterns and improve designation for spatial conservation prioritisation (Fig. 1). In addition, because macroscale species abundance reflects fundamental properties of communities, such as their absolute size and species abundance within geographic units, its estimation can help to advance concepts and frameworks in ecology that have

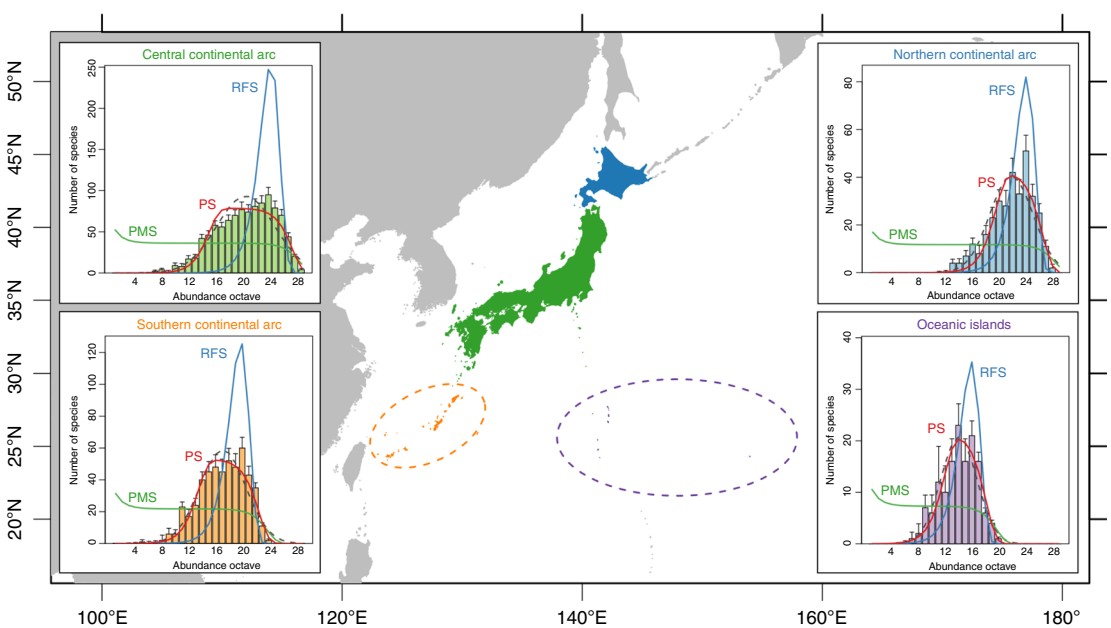

**Fig. 4 Metacommunity species abundance distribution in the four ecoregions of the East Asian islands.** Ecoregions are discerned by colour (central continental arc: green, northern continental arc: blue, southern continental arc: orange, oceanic islands: purple). Histograms in the inner panels represent the estimated metacommunity species abundance distributions (SADs). Error bars indicate standard errors. The coloured lines represent metacommunity SADs predicted by the three variants of the unified neutral theory of biodiversity and biogeography (UNTB) (PMS – point mutation speciation model; RFS – random fission speciation model, PS – protracted speciation model) fitted to the metacommunity SADs. Dashed lines represent fitted Poisson-lognormal models as simple statistical baselines. The $j$th abundance octave is defined as the range of abundance $n$ satisfying $2^{j-1} \le n < 2^{j}$.

**Table 2 Comparison for the fit of three variants of the unified neutral theory of biodiversity and biogeography (UNTB) and a Poisson lognormal model as a simple baseline.**

|  | Central | Northern | Southern | Oceanic |
|---|---|---|---|---|
| AIC |  |  |  |  |
| PMS | 34,706.21 | 11,671.48 | 14,511.53 | 3388.36 |
| RFS | 36,034.85 | 11,391.19 | 14,544.38 | 3298.71 |
| PS | 33,635.67 | 11,150.39 | 13,922.75 | 3214.74 |
| PLN | 33,652.16 | 11,148.78 | 14,009.33 | 3226.23 |
| Akaike weights |  |  |  |  |
| PMS | 0.000 | 0.000 | 0.000 | 0.000 |
| RFS | 0.000 | 0.000 | 0.000 | 0.000 |
| PS | 1.000 | 0.309 | 1.000 | 0.997 |
| PLN | 0.000 | 0.691 | 0.000 | 0.003 |

Models were compared based on their "composite likelihood" suggested by Alonso and McKane[70].
Abbreviations: Central – central continental arc, Northern – northern continental arc, Southern – southern continental arc, Oceanic – oceanic islands, PMS – point mutation speciation model, RFS – random fission speciation model, PS – protracted speciation model, PLN – Poisson lognormal model, AIC – Akaike information criterion.

been mainly based on species pool properties such as $\gamma$-diversity to date[8].

The fitted model illuminated the geographic variation in the occurrence probability and conditional individual density of species. It revealed that both the occurrence probability and conditional individual density vary with AET, a climate-related covariate, and HII, the index for potential human influence. A positive association between AET and occurrence probability of species explains a geographic gradient of species diversity in which the species richness increases toward the warmer, south-western area of Japan (Fig. 2b, see also Supplementary Fig. 2a). On the other hand, a negative association between HII and

occurrence probability, although weaker than that of AET (Supplementary Fig. 3b), explains a tendency of species richness to decrease in areas with higher population density and degree of urbanisation (Fig. 2b and Supplementary Fig. 2b). These results suggest an influence of climate and human activities on the geographic structure of species abundance of woody plants in this region. The mostly opposing partial effect of the covariates and a negative estimate of the correlation coefficient of cell-specific random effects (Supplementary Note 2) indicate a negative covariation of the occurrence probability and individual plant density per species among grid cells. This may be due to a constraint of the total woody plant density at a certain level (Fig. 2a), which can result in a negative correlation between species richness and individual plant density per species.

The results of model validation using independent datasets indicated both positive (in FDP, NFI, and GMFT) and negative (in FSLE) bias of the model prediction. In contrast to these datasets in which individuals greater than a specific size (in their DBH or height) were sampled, or estimated, the present model nominally estimated species abundance of any size by using the vegetation survey data as detection–nondetection observations. In addition, the use of the vegetation survey data is likely to inflate the estimates of individual density, especially for larger species (see Woody plant communities in East Asian islands in the "Methods" section). Hence, we can naturally expect a tendency of overprediction of the model relative to the validation datasets. The observed bias can be explained, however, by the difference in the size of individuals that are considered in each dataset. The negative bias was observed only in the FSLE dataset in which individuals more than 2 m in height were counted, and thus smaller individuals were presumably covered. Accordingly, the model may effectively predict, on average, the density of individuals of some specific size, despite its nominal definition (Supplementary Fig. 5).

An analysis of predictive errors at the species level suggested that, in terms of RMSE and bias, the prediction of individual

**Table 3 Estimates of community abundance, species richness, diversity index, and parameters relevant to the unified neutral theory of biodiversity.**

|  | Central | Northern | Southern | Oceanic |
|---|---|---|---|---|
| **Abundance** |  |  |  |  |
| Total (metacommunity size $J_M$) | $1.65 \times 10^{10}$ | $4.18 \times 10^{9}$ | $3.26 \times 10^{8}$ | $6.39 \times 10^{6}$ |
|  | $[0.11 \times 10^{10}]$ | $[0.55 \times 10^{9}]$ | $[0.30 \times 10^{8}]$ | $[0.54 \times 10^{6}]$ |
| Mean | $4.73 \times 10^{6}$ | $4.24 \times 10^{6}$ | $2.37 \times 10^{6}$ | $3.76 \times 10^{5}$ |
|  | $[0.32 \times 10^{6}]$ | $[0.56 \times 10^{6}]$ | $[0.22 \times 10^{6}]$ | $[0.31 \times 10^{5}]$ |
| Standard deviation | $5.40 \times 10^{6}$ | $5.77 \times 10^{6}$ | $3.64 \times 10^{6}$ | $6.55 \times 10^{5}$ |
|  | $[0.49 \times 10^{6}]$ | $[0.56 \times 10^{6}]$ | $[0.98 \times 10^{6}]$ | $[0.77 \times 10^{5}]$ |
| **Species richness** |  |  |  |  |
| Total ($\gamma$-diversity) | 1024 | 328 | 508 | 141 |
|  | [0.0] | [0.0] | [0.0] | [0.0] |
| Mean ($\alpha$-diversity) | 243.6 | 97.0 | 205.0 | 50.6 |
|  | [0.8] | [0.4] | [0.2] | [0.0] |
| Standard deviation | 74.5 | 32.8 | 76.2 | 33.1 |
|  | [0.3] | [0.2] | [0.2] | [0.0] |
| **Shannon entropy** |  |  |  |  |
| Total ($\gamma$-diversity) | 5.54 | 4.82 | 5.11 | 3.98 |
|  | [0.15] | [0.19] | [0.16] | [0.31] |
| Mean ($\alpha$-diversity) | 4.64 | 4.16 | 4.39 | 2.85 |
|  | [0.13] | [0.16] | [0.13] | [0.15] |
| Standard deviation | 0.36 | 0.33 | 0.42 | 0.91 |
|  | [0.03] | [0.06] | [0.05] | [0.08] |
| **Point mutation speciation model** |  |  |  |  |
| $\theta$ | 52.3 | 16.9 | 31.4 | 10.6 |
|  | [0.3] | [0.1] | [0.4] | [0.3] |
| $\nu$ | $3.18 \times 10^{-9}$ | $4.06 \times 10^{-9}$ | $9.62 \times 10^{-8}$ | $1.65 \times 10^{-6}$ |
|  | $[0.43 \times 10^{-9}]$ | $[0.75 \times 10^{-9}]$ | $[3.75 \times 10^{-8}]$ | $[2.65 \times 10^{-6}]$ |
| $L$ | 19.6 | 19.3 | 16.2 | 13.3 |
|  | [0.1] | [0.2] | [0.2] | [0.3] |
| **Random fission speciation model** |  |  |  |  |
| $\theta$ | 1023.8 | 327.8 | 507.8 | 140.8 |
|  | [0.0] | [0.0] | [0.0] | [0.0] |
| $\nu$ | $3.87 \times 10^{-15}$ | $6.16 \times 10^{-15}$ | $2.42 \times 10^{-12}$ | $4.85 \times 10^{-10}$ |
|  | $[1.37 \times 10^{-15}]$ | $[2.60 \times 10^{-15}]$ | $[3.45 \times 10^{-12}]$ | $[60.3 \times 10^{-10}]$ |
| $L$ | $1.61 \times 10^{7}$ | $1.27 \times 10^{7}$ | $6.43 \times 10^{5}$ | $4.54 \times 10^{4}$ |
|  | $[0.11 \times 10^{7}]$ | $[0.17 \times 10^{7}]$ | $[0.59 \times 10^{5}]$ | $[0.38 \times 10^{4}]$ |
| **Protracted speciation model** |  |  |  |  |
| $\theta$ | 113.5 | 61.8 | 76.6 | 31.7 |
|  | [6.7] | [5.4] | [7.4] | [7.5] |
| $\nu$ | $2.72 \times 10^{-13}$ | $2.78 \times 10^{-14}$ | $3.03 \times 10^{-11}$ | $1.74 \times 10^{-9}$ |
|  | $[0.11 \times 10^{-13}]$ | $[1.20 \times 10^{-14}]$ | $[0.99 \times 10^{-11}]$ | $[12.6 \times 10^{-9}]$ |
| $L$ | $2.19 \times 10^{5}$ | $2.58 \times 10^{6}$ | $4.89 \times 10^{4}$ | $1.21 \times 10^{4}$ |
|  | $[2.77 \times 10^{5}]$ | $[0.63 \times 10^{6}]$ | $[1.45 \times 10^{4}]$ | $[0.10 \times 10^{4}]$ |

Species diversity is represented by Shannon entropy. The abundance estimates nominally encompass individuals of any size, including seedlings. Parameters related to the neutral models are: fundamental biodiversity number $\theta$, speciation rate (per individual per generation) $\nu$, and average species lifetime (generations) $L$. Standard errors are shown in brackets.
Abbreviations: Central – central continental arc, Northern – northern continental arc, Southern – southern continental arc, Oceanic – oceanic islands.

density was in general reasonable for most species with lower density, although it can be considerably biased negatively in some species (Supplementary Fig. 4). This bias can be explained by an extra-Poisson variation in local species abundance, i.e. a violation of the model assumption of a superposed homogeneous Poisson process, in which the model fails to predict extremely high species abundance that arises with lower frequency (see Supplementary Note 2). It occurred more frequently in species with higher density and in those with a narrow geographic distribution; for the latter, the prediction may have tended to be inaccurate given that the number of grid cells within the species range was limited.

Although the individual density of species between the model prediction and validation datasets was positively correlated on a log–log scale, the strength of the association was not substantial (Table 1). A number of potential reasons for the weak correlation can be postulated, including, e.g. dissimilarity in the definition of individuals, the difference in the spatial scale, the high variability of species abundance possibly induced by aggregated distribution, insufficiency of data, and the lack of appropriate covariates and modelling assumption; however, among these, the major factor was not identified in our analysis. In the preliminary model comparison, however, we found that a better explanation for the geographic variation in species abundance, achieved via data integration and the inclusion of covariates and/or correlated random effects, resulted in a better correlation between the model prediction and validation datasets (Supplementary Table 1). The results suggest that further improvement can be expected, at least, in a future analysis with additional model components that explicitly explain variation in abundance over space and species. In addition, we note that the lack of external abundance data that have a spatial scale and size of individuals similar to the present model makes the validation even more difficult. Some large-scale

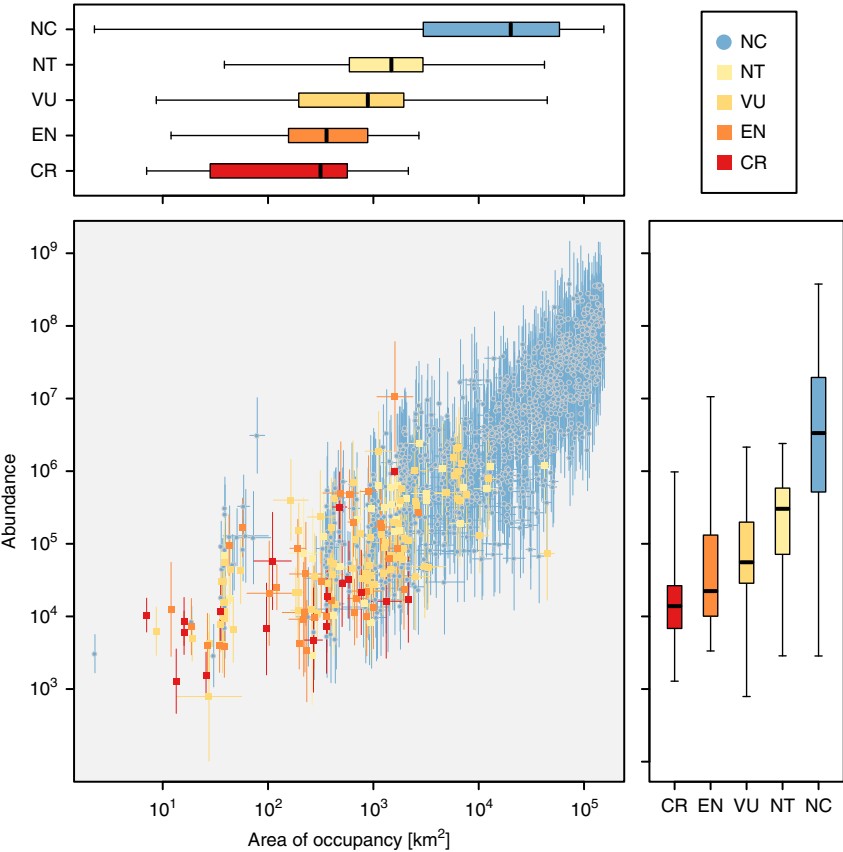

**Fig. 5 Abundance and area of occupancy of woody plant species in natural forests in Japan, with relation to their national red list categories.** NC not classified; NT near threatened; VU vulnerable; EN endangered; CR critically endangered. Error bars in the scatter plot indicate standard errors. Box plots: middle line, median; box, first and third quartiles; whiskers, minima and maxima.

"ground-truthing" data, possibly obtained by using remote-sensing technologies, will likely facilitate further testing to estimate the validity of the macroscale estimates of species abundance.

The UNTB, originally formulated with the point mutation and random fission speciation[5], can fit well to empirical SADs at local communities. However, it has been criticised because of failing to explain the evolutionary aspects such as average species lifetime[23,29,30]. The concept of the protracted speciation achieved a considerable advancement of the UNTB and led to realistic predictions about macroevolutionary patterns of communities[11,31,32]. Nevertheless, in the explanation of empirical SADs, its superiority over the other speciation modes has been unapparent, probably due to limited sample size[11]. Our study fulfills the gap between these theoretical and empirical developments in the UNTB by revealing metacommunity SADs across the four ecoregions in East Asian islands and provides a strong support for the protracted speciation model.

An analysis of metacommunity SADs also highlighted region-specific evolutionary processes, which can shape large-scale biodiversity patterns relevant to geographic characteristics (e.g. area, degree of isolation, and other physiographical conditions) of the regions[33–35]. Greater estimates of the speciation rate in regions of southern continental arc and oceanic islands than in the other two continental arc regions (Table 3) clearly indicate that these regions bear greater species diversity relative to their small land area (i.e. the metacommunity size). They are likely to reflect adaptive/non-adaptive radiation driven by historical vicariance[36,37], which may have led these regions to act as "cradles of biodiversity"[38]. A fundamental limitation in our analysis was, however, that an

immigration of new species realised by a long-distance dispersal from other biogeographic regions cannot be distinguished from an endemic diversification of species, and therefore the estimates of speciation rate represent the joint consequence of these two processes. Long-distance dispersal is another critical macroecological process[39–41], which is especially likely to be promoted in the southern continental arc region by the repeated land bridge connections throughout the Cenozoic. Future studies exploring a further theoretical and methodological development to infer the relative role of speciation and long-distance dispersal are warranted[12].

Although species extinction occurs at the global scale, actions and legislation for conservation are typically conducted at the national scale. In this regard, a number of national or regional red lists have been developed in which categories and criteria for the IUCN Red List are often applied with some modifications[26]. Quantitative information on distribution and abundance of species form the basis of the criteria[13]; assessments of species distribution and abundance over national (or regional) scales are therefore crucial for operative conservation planning. In Japan, the national red list of vascular plants has been compiled based on systematic field surveys of populations comprising about 2100 taxa (including subspecies and variant species) conducted over all prefectures[28,42]. Our result highlighted, however, the possibility that even with such a comprehensive assessment, a number of species may remain unlisted despite their potential threats. In Supplementary Note 4, we provide a list of native species that are not classified to the at-risk categories in the national red list but have rather limited abundance and area of occupancy. Specifically, such species include, for example, *Distylium lepidotum*,

*Neolitsea gilva*, and *Osteomeles schwerinae*, most of which are endemic to the oceanic ecoregion (Fig. 4). Given their regional rarity and high endemicity, these species may deserve higher conservation priority. The incompleteness of information about species population status is one of the major challenges in red listing of species[27]. The statistical framework proposed here may contribute to fill this information gap by providing estimates of species abundance over geographic scales that can facilitate a rigorous assessment for listing and ranking of threatened species, leading to more informed decisions for biodiversity conservation.

Although we showed that a hierarchical modelling approach enables us to estimate macroscale species abundance, a number of limitations remain in practice. First, the model does not account for temporal changes in species abundance. In our application, we used data collected within a period of 60 years (Supplementary Fig. 1c), assuming implicitly that individual density of each species was invariant over this time period. The vegetation would probably have changed, however, at least due to recent environmental changes, such as the increase of herbivorous animals (e.g. deer[43]) and the spread of introduced species[44,45]. Using the dataset for a specific time period may better reflect the species abundance at a certain point of time, but at the expense of reduced sample size. A future modelling effort to accommodate temporal variability of species abundance will, therefore, be useful to fully utilise accumulated data. Second, the inference relies on the ecologically implausible assumption of a superposed homogeneous Poisson point process for the spatial alignment of individuals. A violation of this assumption can lead to an underestimation of species abundance, as seen in our application (Supplementary Note 2), because a spatial clustering of individuals inflates the probability of nondetection of a species within a sampling plot[46,47]. Accounting for such aggregated distribution and the resulting heterogeneity in individual density within a geographic unit should be a key consideration for obtaining better estimates. Finally, the statistical inference may be computationally expensive. To predict individual density that is specific to each species and geographic unit, the model is specified with a large number of unknown quantities (specifically, in the form of random effects). They may render model fitting, model extension, and the associated uncertainty assessment difficult, or even impractical, especially in a specific application with an enormous number of species and geographic units. As such, inference of the macroscale species abundance is an ecologically relevant, yet challenging task that warrants further research.

## Methods

**A framework to estimate macroscale species abundance**. In the following sections, we describe a hierarchical modelling approach that estimates species abundance in discrete geographical units (e.g. grid cells) from spatially replicated multispecies detection–nondetection observations, in combination with various data sources indicating the geographic distribution of species (Fig. 1). A hierarchical model is composed of a series of submodels, including an observation model describing the distribution of data conditional on some latent state variables and a system model describing the variation in the state variables[14–16]. In the following, we first describe a generalised linear mixed model (GLMM), which explains the multispecies detection–nondetection observations in terms of individual density of each species, and therefore explicitly links binary observations to underlying species abundance. Then, we extend this model to incorporate other sources of information about species occurrence that facilitate the inference of abundance for a number of species over a large geographical extent. To describe the basic modelling idea, we here illustrate the models in their simplest form, which includes only intercept and independent random effect terms. The specific, more complex model that we used in our application is described in the later section.

**A model for spatially replicated detection–nondetection data**. We assume that there is a set of geographic areas of interest that contain $I$ species of interest and are divided into $J$ geographical grid cells. Suppose that grid cell $j (j = 1, …, J)$ contains $K_j > 0$ replicated sampling plots in which occurrence was assessed for each species. We denote detection (1) or nondetection (0) of species $i$ in plot $k$ in grid cell $j$ as $y_{ijk}$ ($i = 1, …, I; j = 1, …, J; k = 1, …, K_j$). We also assume that the

area of each sampling plot was recorded, and denote the area of sampling plot $k$ in grid cell $j$ as $a_{jk}$.

The goal of the inference is to estimate the abundance of each species within each grid cell from these locally replicated detection–nondetection observations. To achieve this, we explicitly make several key assumptions in the data generating process. First, we assume that individuals are distributed within some suitable habitats (e.g. forests) in which sampling plots are placed so that they never overlap. The area of suitable habitat is supposed to be known for each grid cell $j$, which we denote as $A_j$. Second, we assume that for each grid cell the spatial point pattern of individuals within the habitats can be regarded as an independent superposition of homogeneous Poisson point processes, each of which represents the spatial alignment of individuals of a species. In the ecological context, this assumption implies that the centres of individuals are regarded as points, and individuals are distributed independently of one another with species-specific individual densities that are constant within a grid cell[48].

These assumptions give us a probability function that explicitly links the probability of species detection within a plot to the density of that species in the grid cell. Let us denote the individual density of species $i$ in grid cell $j$ by $d_{ij}$. Then, the number of individuals occurring in a plot of area $a_{jk}$ independently follows a Poisson distribution with a mean of $d_{ij}a_{jk}$[48]. Therefore, the probability for detecting at least one individual of species $i$ in plot $k$ in grid cell $j$, $p_{ijk}$, can be written as

$$p_{ijk} = 1 - \exp(-d_{ij}a_{jk}) \qquad (1)$$

where $\exp(-d_{ij}a_{jk})$ corresponds to the probability mass of a Poisson distribution with a mean $d_{ij}a_{jk}$ at zero (i.e. a probability that the plot captures no individuals).

On the basis of these settings and assumptions, we provide a state space formulation of the first hierarchical model we consider, in which the model is described in terms of a series of submodels that are conditional on latent state variables and parameters[14–16]. The latent variable of the model was the cell-level individual density of species, which we have already defined as $d_{ij}$.

The observation model describes the occurrence of species within a sampling plot. We can regard the detection–nondetection observation of species, $y_{ijk}$, as a random variable that independently follows a Bernoulli distribution with a detection probability $p_{ijk}$:

$$y_{ijk} \sim \mathrm{Bernoulli}(p_{ijk}), \qquad (2)$$

where $p_{ijk}$ is determined by Eq. (1) under the assumption of the superposed homogeneous Poisson point process.

The system model describes variation in the individual density $d_{ij}$. We decompose the logarithm of $d_{ij}$ into an intercept term $\mu$ and three normally distributed random effects, species $e_i^{(1)}$, grid cell $e_j^{(2)}$, and the combination of species and grid cell $e_{ij}^{(3)}$:

$$\log d_{ij} = \mu + e_i^{(1)} + e_j^{(2)} + e_{ij}^{(3)} \qquad (3)$$

$$e_i^{(1)} \sim \mathcal{N}(0, \sigma_1^2) \qquad (4)$$

$$e_j^{(2)} \sim \mathcal{N}(0, \sigma_2^2) \qquad (5)$$

$$e_{ij}^{(3)} \sim \mathcal{N}(0, \sigma_3^2). \qquad (6)$$

These submodels jointly construct a Bernoulli GLMM with complementary log–log link, in which $a_{jk}$ is treated as an offset term. The model can therefore be fitted to data with standard GLMM packages that implement multiple random effects, such as lme4 in R[49].

We note that, in practice, the size of the geographical units must be determined arbitrarily. Because the model assumes a constant individual density within each unit and the density is determined via replicated detection–nondetection observations, there is a trade-off along the size of the units. Thus, establishing smaller geographical units is likely to increase the likelihood of an inference conforming to the modelling assumption by reducing environmental heterogeneity and hence variation in individual density within the units. However, this comes at the expense of an increased computational burden and a less accurate estimation of individual density. This is because the number of geographical units will be increased for a given spatial extent, and thus the number of random-effect terms will also increase while decreasing the number of replicated plots within the units. Conversely, establishing larger geographical units will ensure a larger number of replications and fewer random effects but will also increase the likelihood of violating the modelling assumptions. Therefore, the size of geographical units should be carefully determined depending on the details of each application.

In addition, it should be remembered that the estimation of individual density relies on the assumption of the superposed homogeneous Poisson point process that was made for modelling convenience rather than for ecological plausibility, and that its violation can result in a biased inference. Specifically, spatial clustering of individuals (i.e. aggregated distribution) may lead to an underestimation of individual density because it inflates the probability of nondetection of species within a sampling plot[46,47]. Even when the assumption of the superposed homogeneous Poisson point process holds, the individual density of species can be underestimated or overestimated when the plots were selectively placed to sample

or avoid specific species. Furthermore, the Poisson point process assumption implies that individual density may not be adequately identified when the individual density or the area of sampling plot, or both, is significantly high because the derived detection probability (Eq. (1)) approaches 1 as $d_{ij}a_{jk}$ increases. This suggests a possibility that, under the random effect formulation where the estimates are shrunk towards the average[15], the underlying individual density and sampling design (i.e. specification of the area of plots) can interact to cause underestimation of abundance, especially for dominant species.

**Integrating cell-level occurrence information**. Because information is shared over species and grid cell by random effects, the model described above can provide estimates of individual density that are specific to each species and cell. However, the estimates may be inaccurate, especially in grid cells where the number of plots is limited and species density is low. Moreover, the model does not explicitly account for the "zero-inflated" nature of species abundance, which assumes that individual density of species is structurally zero in the region outside the species range. To overcome these issues, we extend the model to integrate replicated detection–nondetection observations with data that may directly inform about the cell-level presence–absence of species, such as species occurrence records and expert range maps.

We introduce a latent indicator state variable that represents the cell-level presence–absence of species and is denoted as $z_{ij}$. The detection probability $p_{ijk}$ is then expressed as follows:

$$p_{ijk} = 1 - \exp(-z_{ij}d_{ij}a_{jk}), \tag{7}$$

which indicates that the detection probability is 0 when the species is absent in the cell ($z_{ij} = 0$), but it takes $1 - \exp(-d_{ij}a_{jk})$ when the species is present in the cell ($z_{ij} = 1$). Hence, $d_{ij}$ now represents the individual density that is conditional on the presence of that species.

We regard $z_{ij}$ as a random variable following a Bernoulli distribution and add an additional system model component to describe it. By adopting a similar modelling approach applied for the conditional individual density (Eqs. (3)–(6)), the additional components can be constructed as follows:

$$z_{ij} \sim \text{Bernoulli}(\psi_{ij}) \tag{8}$$

$$\text{logit } \psi_{ij} = \eta + u_i^{(1)} + u_j^{(2)} \tag{9}$$

$$u_i^{(1)} \sim \mathcal{N}(0, \tau_1^2) \tag{10}$$

$$u_j^{(2)} \sim \mathcal{N}(0, \tau_2^2), \tag{11}$$

where $\psi_{ij}$ is the occurrence probability of species $i$ in cell $j$, which was decomposed into an intercept term $\eta$ and two normally distributed random effects that vary over species $u_i^{(1)}$ and cells $u_j^{(2)}$ on a logit scale.

We assume that the cell-level species occurrence $z_{ij}$ is partially observed via the plot-level detection–nondetection observations and/or the auxiliary cell-level presence–absence information that is independent of the detection–nondetection observations. A cell-level presence of species may be registered, for example, by museum-based or herbarium-based specimens and/or occurrence records, while absence of species may be deduced by exploiting, for example, expert range maps[50] and/or regional species checklists. In general, the information about the species absence should be treated conservatively because it is difficult to verify[50]; therefore, the evidence of species presence should be prioritised when different sources of data are in conflict.

Technical details of the inference of this integrated model are described in Supplementary Note 1. We note that this model even enables us to utilize geographical grid cells that contain no detection–nondetection observations but have cell-level presence–absence information for some species (see Supplementary Note 1).

**Ecological variables obtained as derived quantities**. Once estimates (or posterior samples, in case of fully Bayesian approach) of random effects are obtained, we can derive the estimates of $\psi_{ij}$ and $d_{ij}$, denoted by $\hat{\psi}_{ij}$ and $\hat{d}_{ij}$, respectively, by substituting the estimates of random effects into Eqs. (9) and (3), respectively. Based on these estimates, we can further derive estimates for a wide array of variables that are of ecological interest. For example, let us use $m_{ij} = 1$ to denote that the value of $z_{ij}$ is known (through either the detection–nondetection observations or the cell-level occurrence information) for species $i$ in cell $j$ and $m_{ij} = 0$ denotes otherwise. Then, the number of modelled species that are actually present in cell $j$, denoted by $S_j$, can be estimated as

$$\hat{S}_j = \sum_{i=1}^{I} \left\{ m_{ij}z_{ij} + (1 - m_{ij})\hat{\psi}_{ij} \right\}. \tag{12}$$

Note that the use of the estimated occurrence probabilities $\hat{\psi}$ enables this estimator to account for the possibility of the presence of species even when they are not detected in the replicated plots (cf. ref. [51]) or no detection–nondetection observation is available in the grid cell. Let $\mathbf{N}_j$ denotes the vector of abundance of all

species in cell $j$, the property of an ecological community that we aimed to infer, and can be estimated for each grid cell as

$$\hat{\mathbf{N}}_j = \left\{ \hat{d}_{ij}A_j \left[ m_{ij}z_{ij} + (1 - m_{ij})\hat{\psi}_{ij} \right] \right\}_{1 \le i \le I}, \tag{13}$$

where as defined above, $A_j$ denotes the area of habitats in cell $j$. We can also estimate the species abundance for a subset of the area of interest $\mathcal{J}$, denoted by $\mathbf{N}_{\mathcal{J}}^*$, as follows:

$$\hat{\mathbf{N}}_{\mathcal{J}}^* = \left\{ \sum_{j \in \mathcal{J}} \hat{d}_{ij}A_j \left[ m_{ij}z_{ij} + (1 - m_{ij})\hat{\psi}_{ij} \right] \right\}_{1 \le i \le I}. \tag{14}$$

Note that the estimates of abundance of each species further permit to obtain various diversity indices that are a function of a vector of (relative) abundance, such as Shannon entropy and Gini–Simpson index, as well as other generalised metrics including phylogenetic/functional diversity indices and the Hill numbers[52]. In addition, the total area of occupancy within the modelled geographic areas, denoted by $R$, can be estimated for each species as follows:

$$\hat{R}_i = \sum_{j=1}^{J} A_j \left[ m_{ij}z_{ij} + (1 - m_{ij})\hat{\psi}_{ij} \right]. \tag{15}$$

**Model variants**. The above model has the most trivial form in the sense that the variation in individual density was explained only by several unstructured random-effect components that share information across all grid cells and species[53–55]. It can be extended, however, in various ways to accommodate additional complexities. For example, in an analogous fashion to many other classes of hierarchical models and species distribution models (SDMs), environmental covariates could be introduced in the system model to explicitly describe the association between environmental factors and individual density. The model could also explain the correlation structure of random effects on the geographic and/or phylogenetic space in an explicit manner[56,57]. Additionally, the correlation between random effects for conditional density ($e$) and that for occurrence probability ($u$) could be accounted for, as shown in the following application. Such generalisations will potentially enhance the model prediction and provide further ecological insights. Nevertheless, some of these extensions may be difficult to adopt in practice, especially in studies that examine a very large number of species and grid cells, as is the case with our application described below, because the model may involve an excessive number of parameters and/or a huge covariance matrix, rendering the inference computationally challenging[54].

**Woody plant communities in East Asian islands**. We applied the proposed framework to a dataset of woody plant communities in mid-latitude forests in Japan. For the replicated detection–nondetection observations, we compiled a large dataset from vegetation surveys that consists of 40,516 georeferenced plots which comprises the dataset of Kusumoto et al.[58] and the national vegetation survey of Japan (http://www.biodic.go.jp/english/kiso/vg/vg_kiso_e.html). These plots were placed in natural forests in various successional stages between 24°02′–45°30′N and 122°56′–153°59′E (Supplementary Fig. 1a). The plot area ranged from 0.01 to 18,000 m² where the typical size was about 100 m² (Supplementary Fig. 1b). The time period of the surveys spans from 1954 to 2013, in which a substantial proportion of data have been collected between the 1970s–1980s and 2000s–2010s (Supplementary Fig. 1c).

In the vegetation survey, species occurrence in the sampling plots (called "relevés") is traditionally recorded according to cover classes for individual species at every different layer. We converted these vegetation observations into detection–nondetection records by assigning 1 if the species appeared in the plot and 0 otherwise. In this analysis, we standardised the names of woody plant species and pooled the data for varieties and subspecies with those of their parent species following a list of Japanese plant names[59]. As a result, we obtained detection–nondetection observations for 1248 species, which includes introduced species and covers almost every woody plant species found in Japan.

It should be noted that as the survey is based on the measurement of plant cover, species occurrence can be recorded regardless of the size of individuals within a plot. This implies that in the following analyses, estimates of species abundance nominally encompass individuals of any size, including seedlings (but not seeds), that are observable and identifiable during the survey. In addition, in the vegetation survey, species can be recorded even when their stems are absent from the plot, and some branches are hanging over the plot. In terms of the Poisson point process assumption of the model, this implies that the area of the plot is effectively enlarged, so that some individuals located outside the plot can be detected. Therefore, the use of the cover data can lead to a positive bias in the inference of species abundance, especially for large tree species. Such a bias may become less relevant, however, in plots with a larger area, which have smaller margins relative to their area.

We divided the entire study area into $10 \times 10$ km grid cells[37,60]. We analysed in total 4619 cells, which covered ca. 99.2% of the total land area of Japan. In total 3683 cells contained at least one vegetation plot.

We also compiled the species occurrence information at a cell level based on multiple data sources that were independent of the vegetation survey dataset.

Species presence was registered from museum and herbarium specimens, species occurrence records, and distribution maps of plant species compiled by Horikawa[61]. Species absence was recorded from the distribution maps[61] and regional species checklists compiled by prefectures of Japan[36].

**Model fitting and inference**. As a preliminary analysis, several model variants were fitted to these data using the maximum marginal likelihood procedure and then compared based on AIC and three other benchmarks of predictive performance that we described below. The preliminary models included a model without data integration, a model with data integration, a model with correlated random effects, and several models with covariate effects. Details of this preliminary analysis are described in Supplementary Note 2. As a result, an integrated model with two cell-specific covariates, AET, and the HII[62], along with their interaction effect and correlated random cell effects, was selected as the best model and used in the following analyses. Specifically, the model is specified by a series of equations as follows:

$$y_{ijk} \sim \text{Bernoulli}(p_{ijk}) \tag{16a}$$

$$p_{ijk} = 1 - \exp(-z_{ij} d_{ij} a_{jk}) \tag{16b}$$

$$z_{ij} \sim \text{Bernoulli}(\psi_{ij}) \tag{16c}$$

$$\log d_{ij} = \mu + \beta_1 x_{1j} + \beta_2 x_{2j} + \beta_3 x_{1j} x_{2j} + e_i^{(1)} + e_j^{(2)} + e_{ij}^{(3)} \tag{16d}$$

$$\text{logit } \psi_{ij} = \eta + \gamma_1 x_{1j} + \gamma_2 x_{2j} + \gamma_3 x_{1j} x_{2j} + u_i^{(1)} + u_j^{(2)} \tag{16e}$$

$$\begin{pmatrix} e_i^{(1)} \\ u_i^{(1)} \end{pmatrix} \sim \mathcal{N}_2 \left( \begin{bmatrix} 0 \\ 0 \end{bmatrix}, \begin{bmatrix} \sigma_1^2 & 0 \\ 0 & \tau_1^2 \end{bmatrix} \right) \tag{16f}$$

$$\begin{pmatrix} e_j^{(2)} \\ u_j^{(2)} \end{pmatrix} \sim \mathcal{N}_2 \left( \begin{bmatrix} 0 \\ 0 \end{bmatrix}, \begin{bmatrix} \sigma_2^2 & \rho \sigma_2 \tau_2 \\ \rho \sigma_2 \tau_2 & \tau_2^2 \end{bmatrix} \right) \tag{16g}$$

$$e_{ij}^{(3)} \sim \mathcal{N}(0, \sigma_3^2), \tag{16h}$$

where $x_{1j}$ and $x_{2j}$, respectively, represent the values of AET and HII in cell $j$, scaled to have mean 0 and variance 1. Note that the correlation between the two cell random effects, $e_j^{(2)}$ and $u_j^{(2)}$, is accounted for by specifying a bivariate normal distribution with an additional correlation parameter $\rho$. A 10-fold cross-validation was conducted to asses the goodness of fit of the model to replicated detection–nondetection observations in which the mean area under the curve (AUC) for validation datasets was estimated as 0.894 (standard deviation: 0.001). To calculate the AUC, we omitted data where the absence of species was indicated by the auxiliary species occurrence information to exclude trivial predictions for structural zeros.

Based on the estimates of the selected model, the abundance and area of occupancy of 1248 woody plant species within natural forests was estimated for 4619 grid cells by using Eqs. (13) and (15), respectively. The area of natural forest in each cell was obtained based on the national survey of the natural environment (http://www.biodic.go.jp/trialSystem/EN/info/vg.html).

The uncertainty associated with the estimates of species abundance, area of occupancy, and quantities obtained as a function of these were evaluated by using the parametric bootstrap procedure for hierarchical models[63] in which 100 bootstrap samples were obtained to measure the standard error of each estimate. To accommodate reduced uncertainty in the geographic distribution of each species (i.e. $z_{ij}$) due to the availability of the auxiliary cell-level presence–absence information, the bootstrap procedure was conducted with the auxiliary dataset being fixed.

**Validation**. The estimates of species-specific individual density were validated based on data from geographically replicated forest inventory plots that were independent of the fitted data. We used three sources of forest inventory data that were collected in natural forests in Japan. They include the forest dynamics plots (FDP), the national forest inventory plots (NFI), and forest sampling plots along latitudinal and elevational gradients (FSLE). Sampling procedures and spatial coverage differed between the inventory data as we explain below.

The FDP dataset consists of species abundance data collected from 40 quadrats. In each quadrat, which was usually 1 ha in size, individuals with a diameter of approximately >5 cm at breast height (DBH) were monitored[64]. This dataset fairly represents the mosaic structure of forests with different developmental stages and thus is expected to precisely capture local population size for common climax species in old growth mountain forests, while it may poorly represent the population of pioneer or fugitive species, especially in lowland forests. The FDP dataset is publicly available from the Biodiversity Center of Japan (http://www.biodic.go.jp/moni1000/findings/data/index_file.html).

The NFI dataset included 7672 plots in which woody plant individuals were assessed in nested concentric circular plots. Individuals with DBH >1 cm were

measured in a 0.01 ha circular area, while those with DBH >5 and >18 cm were surveyed in a 0.04 and 0.1 ha circle, respectively (http://www.rinya.maff.go.jp/j/keikaku/tayouseichousa/). The NFI plots were systematically placed in a 4 km × 4 km grid laid over entire Japan and thus were expected to provide less-biased samples of density of woody plants. The NFI dataset is publicly available from the Forestry Agency of Japan (http://www.rinya.maff.go.jp/j/keikaku/tayouseichousa/chousadeta.html).

The FSLE dataset included 460 plots placed within climax forests in five regions (central and western Hokkaido, Shizuoka, Miyazaki, and Kagoshima prefectures) in Japan (unpublished data by Y. Kubota). For each region, abundances of woody plants were surveyed along an elevational gradient in which 10 replicated plots were located for each 100–200 m elevation interval. For each plot, woody plant individuals larger than 2 m in height were counted over a 0.01 ha area. The sampling design along both the elevational and latitudinal gradients was expected to well reflect the environmental heterogeneity in the mid-latitude forests.

For each dataset, the species abundance data were pooled within each grid cell and compared to the model prediction to estimate root mean square error and bias in the prediction of individual density, in addition to the correlation between observation and prediction on a log–log scale. For each grid cell, observed and predicted individual density were compared for all species, except for the species whose absence in the cell is indicated by the cell-level occurrence information. In order to predict the abundance in NFI plots, we set the area of each plot to 0.1 ha.

Additionally, prediction of the total individual density was validated with a global estimate of tree density[17], which here we call the global map of forest trees (GMFT). The GMFT provides the density of trees with DBH >10 cm at 1-km[2] resolution that was predicted based on regression models for forested areas in 14 biomes[17]. The GMFT estimates are provided in the raster format and are publicly available (https://elischolar.library.yale.edu/yale_fes_data/1/). The GMFT estimates were matched to the grid specification adopted in this study to obtain the individual density per grid cell, which was then converted to the individual density per 1 km[2] natural forest by dividing with the area of natural forest for each cell. Because GMFT does not distinguish natural and planted forests, the GMFT estimates were scaled for each cell by the proportion of the area of natural forest to that of the entire forest (i.e. both natural and planted forests) within the cell to correct GMFT estimates for natural forests. The area of planted forest in each cell was obtained based on the national survey of the natural environment (http://www.biodic.go.jp/trialSystem/EN/info/vg.html). The corrected GMFT estimates were used to validate the prediction of individual density in each cell, in the same manner as the validation with forest inventory data; they were also used to obtain the total woody plant abundance within the natural forests in the entire region. We note that although there was an alternative GMFT that was based on regression models in 813 ecoregions, the results of validation were quantitatively similar to that with the biome models.

Additional details of the validation can be found in Supplementary Note 2.

**Inference of macroevolutionary processes in metacommunities**. The UNTB[5] provides a mechanistic explanation of the origin and maintenance of biodiversity; based on the premise that all individuals in a system are functionally equivalent and thus follow neutral processes of demography, dispersal, and speciation, the UNTB derives SADs, at both local-community and meta-community (i.e. species pool) scales, in addition to a range of other macroecological and macroevolutionary patterns such as the species–area relationship[40], β diversity[65], and various phylogeny characteristics[66].

The UNTB bridges evolutionary biology and community ecology by linking, theoretically, macroevolutionary processes to biodiversity patterns. In particular, it predicts that the statistical form of the SAD in the metacommunity is dependent on the mode of speciation[5,11,12,18–20]. The point mutation speciation model, which formed the basis of the first UNTB proposed by Hubbell[5], models speciation as a process in which each new species is represented initially by a single individual. The point mutation speciation model predicts a metacommunity SAD that follows the logseries distribution, a distribution that is characterised by a relatively high proportion of rare species[5,21]. In contrast, the random fission speciation model assumes that speciation occurs in the metacommunity owing to the random division of a population of an existing species[12,19]. The random fission speciation model predicts a fairly even metacommunity structure, which is related to the MacArthur's[67] broken-stick model[12,19]. The point mutation speciation and random fission speciation models represent the two extremes of a spectrum of speciation modes in UNTB. This spectrum of speciation modes has been argued to be unified with the concept of protracted speciation, which characterises speciation as a gradual, drawn-out process[11,20]. The UNTB with protracted speciation predicts a metacommunity SAD that follows a difference-logseries distribution. The difference-logseries distribution follows a logseries distribution at large abundances, while behaving differently at small abundances; namely, it predicts fewer rare species than the logseries distribution[11,20].

Our ability to infer evolutionary processes that underpin observed biodiversity patterns is, however, fundamentally limited because species abundance data can be obtained only from local communities. Indeed, earlier studies have shown that differences in the mode of speciation are hardly discerned based on samples from local communities as they may not leave a signature on SADs realised in dispersal-limited localities[5,11,12,18]. The limitation in data acquisition also prohibits us from

identifying the rate of speciation ($v$) from SADs because local community SADs are determined by the fundamental biodiversity number ($\theta$), which is a compound parameter depending both on $v$ and the metacommunity size ($J_M$)[12,21]; but see ref. [18]. Consequently, fundamental macroevolutionary properties of a metacommunity, such as $v$ and the average lifespan of the species ($L$[29]), have remained largely unknown.

Although defining a metacommunity is difficult in practice, discerned biogeographic divisions will approximate its theoretical definition, as they can be regarded an evolutionary unit within which most member species spend their entire evolutionary lifetimes[68]. Based on a previous biogeographic assessment of woody plants in the Japanese archipelago[36] and Takhtajan's floristic provinces[69], we thus divided the archipelago into four ecoregions that belong to different biogeographic divisions (Fig. 4) and obtained metacommunity SADs by aggregating abundance estimates over grid cells within each ecoregion (Eq. (14)). The four ecoregions are defined as follows: (1) The central continental arc region is the largest ecoregion, which includes the three largest islands in Japan (Honshu, Shikoku, and Kyusyu). It encompasses deciduous and evergreen broad-leaved forests and belongs to the Takhtajan's Japan–Korea province; 3478 geographical grid cells belong to this ecoregion. (2) The northern continental arc region is the second largest ecoregion, and it includes Hokkaido, the second largest island of Japan. It encompasses coniferous and deciduous broad-leaved forests and belongs to the Takhtajan's Sakhalin–Hokkaido province. The Tsugaru Strait separates the central continental arc region and northern continental arc region; 986 geographical grid cells belong to this ecoregion. (3) The southern continental arc region is composed of the Nansei Islands and separated from the central region by the Tokara Strait. It encompasses evergreen broad-leaved forests and belongs to the Takhtajan's Tokara–Okinawa province. This ecoregion comprises 138 geographical grid cells. (4) The oceanic islands region is composed of the Bonin (Ogasawara) Islands. It encompasses evergreen broad-leaved forests and belongs to the Takhtajan's Volcano-Bonin province. Differing from other ecoregions, in which almost all the lands are continental islands, the oceanic region is composed of oceanic islands only. It includes 17 geographical grid cells.

For each ecoregion, we fitted and compared three variants of the UNTB to the estimate of the metacommunity SAD. The fitted model includes the point mutation speciation model[5,21], the random fission speciation model[12], and the protracted speciation model[11]; for each of these models, a probability function of the metacommunity species abundance vector (i.e. likelihood function for metacommunity SAD) and/or an analytical solution of the SAD in the stationary metacommunity has been obtained and can be used for model fitting.

The point mutation speciation model was fitted to the metacommunity SADs by using maximum-likelihood method. The likelihood function for metacommunity SAD (i.e. assuming no dispersal limitation) under point mutation speciation model is known as the Ewens sampling formula (e.g. Eq. (2) in ref. [21]). Formal likelihood-based inferences were, however, difficult to obtain for the other two models. Although a sampling formula has been acquired for a metacommunity under random fission models (Eq. (38) in ref. [12]), we were not able to apply this formula to our specific data as it underflows when the size of metacommunity is large, even when high precision arithmetic is used. We thus reached a compromise to use Eq. (21) in ref. [12], which was derived without considering the sampling process, but provides the equilibrium probability function of the species abundance vector in a metacommunity with a fixed size $J_M$. For protracted speciation model, no likelihood function was available. To fit this model, Rosindell et al.[11] used "composite likelihood" that was suggested by Alonso and McKane[70]. This approach was however not practical in our case because of the large metacommunity size, thereby requiring an excess number of evaluations of the expected number of species with specific abundance. This prohibited its adoption in the numerical optimisation procedure. We therefore applied a least-square method to the Preston's abundance octaves of metacommunities. We note that in addition to these three models, we also fitted the per-species speciation model of Etienne et al.[18]. However, this model consistently yielded boundary estimates that made the model identical to the point mutation speciation model. We thus omitted it from the comparison.

These differences in the fitting procedure render the model comparison complicated. To compare fitting of the models, while accounting for differences in the number of parameters (we note that point mutation model and random fission model have one free parameter ($\theta$) but protracted speciation model has two ($\theta$ and $\beta$)), we prefer to use information criteria[9,71] which relies on the formal maximum-likelihood inference[72]. However, to fully utilise this approach was impossible in our application because the likelihood function was available only in the point mutation model. Thus, we compared the models based on the Akaike information criterion (AIC) and the Akaike weights[73] calculated with "composite likelihood"[70], assuming that the parameter estimates of the random fission speciation model and protracted speciation model attain the maximum likelihood. In the model comparison, we also included a Poisson-lognormal mixture model[74] as a flexible, simple baseline statistical model[9].

The objective function (i.e. negative log-likelihood or sum of squared error) of the variants of UNTB was minimised in terms of fundamental biodiversity number $\theta$ (in addition to $\beta$, in the case of protracted speciation model). Estimates of the speciation rate $v$ (per individual per generation) and mean species lifetime $L$ (generations) were then derived as a function of these estimated parameters and metacommunity size $J_M$ (individuals). In point mutation model, $v$ relates to other quantities as $\theta = \frac{v}{1-v}(J_M - 1)$[21], whereas in random fission model, the

relationship is given as $\theta = \sqrt{v}J_M$[12]. In the protracted speciation model, the corresponding equation is given as $\theta = \frac{\mu}{1-\mu}(J_M - 1)$, where $\mu = (1 + \tau)v$ and $\tau = \frac{J_M - 1}{\beta} - 1$[11]. The average species lifetime is obtained from the general equation of Ricklefs[29]: $L = \frac{\text{equilibrium number of species in metacommunity}}{\text{rate of production of new species}}$. The corresponding formula is as follows: for point mutation speciation model, $L \approx -\log v$; for random fission speciation model, $L \approx v^{-\frac{1}{2}}$[12]; for protracted speciation model, $L \approx -\tau \log \tau \mu$[11].

**National red list categories of species in Japan**. We categorised each woody plant species based on the 2019 version of the national red list of Japan. The national red list of Japan adopts categories of threatened species modified from the IUCN Red List (http://www.biodic.go.jp/english/rdb/rdb_f.html). Specifically, the current national red list classifies species into categories compatible with the IUCN Red List (CR, EN, VU, NT, and DD) when they are judged to be at risk of extinction, although it does not have specific categories that correspond to Least Concern (LC) and Not Evaluated (NE) in the IUCN Red List. These two statuses are therefore not distinguishable in the national red list. For this reason, we pooled all species with such status (i.e. species that could not be classified to the at-risk categories), which were collectively labelled as "not classified (NC)".

In the analysis described in Implications for biodiversity conservation, we omitted 130 introduced species that are outside the scope of the national red list. In addition, we ignored red list categories of subspecies or variant species because we have aggregated these taxa to their parent species in the preceding model fitting procedure (see Woody plant communities in East Asian islands).

**Reporting summary**. Further information on research design is available in the Nature Research Reporting Summary linked to this article.

## Data availability
The datasets of vegetation survey and species geographic distribution, that were analysed in the current study, are available at https://www.givd.info/ID/AS-JP-002 upon request from the Global Index of Vegetation-Plot Databases (GIVD). See the GIVD rules for details at: https://www.givd.info/index.xhtml. Other data sources include the national survey of the natural environment (http://www.biodic.go.jp/trialSystem/EN/info/vg.html), the forest dynamics plots (FDP: http://www.biodic.go.jp/moni1000/findings/data/index_file.html), the national forest inventory plots (NFI: https://www.rinya.maff.go.jp/j/keikaku/tayouseichousa/chousadeta.html), and the global map of forest trees (GMFT: https://elischolar.library.yale.edu/yale_fes_data/1/). The forest sampling plots along latitudinal and elevational gradients (FSLE) dataset is unpublished data available upon reasonable request.

## Code availability
The Template Model Builder code for the fitted model is available in the supplementary files (Supplementary Software) and from GitHub (https://github.com/fukayak/mSAD_TMB).

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

## Acknowledgements

We thank S. Eguchi and O. Komori for their helpful comments and discussion. We are grateful to T.J. Matthews for valuable comments and editing. We are particularly grateful to local botanists, vegetation researchers, and naturalists who have accumulated the information on plant distribution through their fieldwork steadies over the past decades. This research was supported by an allocation of computing resources of the SGI ICE X and SGI UV 2000 supercomputers from the Institute of Statistical Mathematics and SGI UV 2000 supercomputers from Institute for Chemical Research, Kyoto University. Financial support was provided by the Japan Society for the Promotion of Science (no. 15H04424), the Environment Research and Technology Development fund of the Ministry of the Environment, Japan (4-1501 and 4-1802), and Program for Advancing Strategic International Networks to Accelerate the Circulation of Talented Researchers, Japan Society for the Promotion of Science.

## Author contributions

Y.K. conceived the ideas; B.K. and T.S. compiled the data; K.F. designed the methodology and conducted data analyses; J.F. contributed to data interpretation and model development; K.F. and Y.K. coordinated the writing of the manuscript. All authors discussed the results and contributed critically to the drafts.

## Competing interests

The authors declare no competing interests.
