## [Peer Review File · Nature Communications]

Reviewers' Comments:

Reviewer #1:

Remarks to the Author:

The authors use spatially replicated multispecies detection-non-detection data, along with supplementary information on species occurrences in the Japanese archipelago, to infer geographic woody plant species abundances in natural forests at the 10 x 10 km scale of resolution. Multi-species distribution distributions (SADs) for trees are constructed from the data. The constructed SADs are then used to determine the most likely sort of macroevolutionary speciation that has occurred, which showed that protracted speciation (versus point mutation and fission) is the most likely process. The authors were also able to estimate speciation rates in four biogeographical region. In addition, the authors show from the SADs that it is possible that some woody plant species not currently on the Japanese red list may be vulnerable to extinction.

The authors' analysis addresses a key problem raised by the unified neutral theory of biodiversity and biogeography (UNTB), that the SAD is sensitive to the mode of speciation (e.g., Rosindell et al. 2010, Etienne and Haegeman 2011). The current paper answers the need, expressed in those papers, for getting beyond local communities to macroscale metacommunity data sets to adequately address the question of what mode of speciation is best, and clearly protracted speciation is able to fit the data very well (Figure 4) and predict reasonable values of metacommunity size, fundamental biodiversity number, speciation rate, and average species lifetimes for different biogeographic regions (Table 2). This seems to be a nice contribution to biodiversity theory.

The use of the authors' SADs to generate data on species at the range of low abundances and small areas of occupancy, is a useful step towards determining if the species in this range that are not currently classified, might need to be listed if further conditions (e.g., species are declining in abundance or habitat fragmentation is occurring) are met.

I have a few comments.

Table 2. The units of speciation rate, v , need to be given. These are 'probability of speciation event per birth in the metacommunity' (page 114, Hubbell 2001).

The methodology of estimating macroscale SADs from spatially replicated detection-nondetection observations using a hierarchical model to link probability of species detection within a plot to density of species in grid cell, described on pages 14-21, is well founded on Royle and Dorazio (2008), so I believe it is basically correct and is easier than doing counts. However, I am wondering about possible underestimation of abundance of the species that are superabundant. If only detection is used to infer abundance, it seems possible that even a large number of replicate plots may not be able to discern the difference between a species that is superabundant on 50% of plots and one that occurs in much lower numbers on the same 50% of plots. Figure 3 seems to indicate that species with higher abundance are more likely to have their abundance underpredicted by the model. For particular cases of underprediction this could be checked. (The authors suggest another reason for underprediction based on assumption of homogeneous Poisson spatial distribution, which also seems reasonable.)

I would like to know more of the details of the 40,000 vegetation surveys. Was the tree size (> 5 cm DBH) used for determining occupancy, as in the FDP data used for validation, or were smaller trees also included? Also, if forest with closed canopy was predominately sampled, early successional species could be missed.

Page 22. The plot sizes range from 0.01 m² to 18,000 m² (page 22), which is an enormous

difference. I think there is a typo and the authors meant 0.01 ha. Even so, the difference is large. Is this likely to create any sampling biases?

Also, it would be good to know what time period the surveys were done. It is not likely that species compositions changed too much over the period of the surveys, but it is useful to know the time period in any case. Perhaps more importantly, species occurrence data were also used from other sources, such as museum and herbarium specimens and early distribution maps (Lines 429-433). Inclusion of these old records carries some risk.

False positives could also occur through misidentification. Is there some quality control on the data that attempt to avoid that?

Are only native woody plants included, versus invasive woody plants that have become established?

Validation is well described on Page 23, and Figure 3 gives a useful summary, but it would be interesting to know if there were particular species that were very poorly estimated, and perhaps why.

Minor comments:

Line 60. Change 'comprised of' to 'comprising'

Line 64. Change 'abundance' to 'abundances'

Line 65. Change 'estimated to' to 'estimated to be'

Line 81. I am not sure what 'a more refined pattern' means here.

Figure 3 caption. Change 'crossed lines' to 'diagonal lines'

Line 141. Change delete 'are'

Line 314. Insert comma after 'inaccurate'

Line 340. Change 'denotes' to 'denote'

Line 515. Change 'proximate' to 'approximate' and insert comma after 'definition'

Line 540. Change 'for these' to 'for each of these'

Reviewer #2:

Remarks to the Author:

The present manuscript presents a novel approach to tackle several fundamental issues of spatial ecology: Where species live, how abundant they are, and can we derive particularly useful conclusions from their distributions and abundance. Not only does the work address all of these simultaneously, it also offers a clear methodological advancement by integrating everything in a single and elegant statistical model. To add further novelty, the analysis/model also explicitly operates at several spatial scales. Finally, the dataset (Japanese trees) is large and solid, and adds robustness and credibility.

Apart from the clear methodological and conceptual advancement, there are some highly relevant and striking results. The first useful result is the suite of maps of abundance for each species (they should definitely be provided with the paper). On top of that, the authors infer, from the estimated large-scale SAD, an elegant and straightforward support for the protracted mode of speciation, and they predict that there are species whose conservation status should be revisited. Thus, their approach and results are relevant for both basic macroecology and macroevolutions, as well as for applied conservation.

Although this seems like a lot to be offered by a single study, it all fits together naturally, mostly because of everything actually follows from a single statistical model. The manuscript is also very well written, I rarely get the chance to review something that well crafted. Finally, I applaud the clarity of

the methods description and the clear mathematical notation. Overall, I think that this can become a milestone and highly cited reference in the research of large scale biodiversity patterns and data integration. As such it is potentially suitable for Nature Communications.

I have, however, some reservations that prevent me from recommending the ms for publication in its current form.

MAJOR COMMENTS

(1) I am really surprised that the authors haven't considered a single environmental predictor in their model. It is really well established that tree distributions, species richness, biomass, or densities can be predicted by variables such as NPP, AET, biomass, NDVI, or some combination of precipitation- and temperature-related climatic variables. Further, these variables are really easy to get at some good resolutions such as 1 x 1 km or even 250 x 250 m – extracting them for the plot or grid data is a matter of one day with raster package in R and function "extract". On lines 289-297 the authors give some justification for why they haven't included this, but the arguments are not very convincing. The environmental predictors don't have to have a specific parameter for each species – they can operate at a higher hierarchical level, e.g. affecting the total density of all species, thus increasing the total number of parameters by just a couple. Moreover, it can even be that if the right predictor is included, both the parameter identifiability and predictive power can improve! I've seen MCMC algorithms in Stan that converged faster (i.e. a single MCMC iteration was actually faster) for more complex models because the models were simply more reasonable.

For a journal such as Nature Communications I would expect the study not to have such an obvious omission. Thus, I strongly suggest that the authors try to add at least 1 or 2 environmental (climate-related or productivity-related) predictors to some part of the model (e.g. predicting d_{ij}). If it is too computationally demanding to use the predictor on a per-species basis, then its effect could be modeled on some higher level, e.g. as a predictor of the total N of all individuals of all species in a cell (or plot, or both). I am aware that this is extra work, potentially protracting the publication process, but it would make a difference between a regular specialist macroecological analysis, and a top-level Nature Communications study. And the current predictive power of the model really asks for improvement, and the role of climate and productivity is just too notorious to be ignored. I am also aware that there may be some problems with this idea, and that you will possibly find out that the model is still better off without such predictors, but I'd like you to at least try.

(2) Along similar lines, I miss at least a mention of the role of human impact on the Japanese forests, and on the parameter estimates and predictions. I would expect this to be really important in a country with high human population density such as Japan. Ideally, this could also be incorporated to the model as a predictor, but I am aware that this could really be too much (although the GIS layers are there). At least it should be discussed.

(3) Would spatial autocorrelation be hard to incorporate? And which part of the model would be the best to incorporate it? Could actually improve the predictions? Again, I see this as optional, but it should at least be mentioned.

(4) The model validation (Fig. 3) shows that the predictive power at local scale is not that great. It may not be such a big problem since the main inference is done on the regional scale. Anyway, I can see that an additional validation on a larger scale can be made by comparing the total predicted abundance at the 10 x 10 km cells with estimates from other studies, e.g. from Crowther et al. This should be an easy extra plot to add – you would just aggregate the values from Crowther over the 10 x 10 km grid cells. In addition, it would be good to have the relationship mentioned on line 82 shown graphically, in a scatterplot. Finally, would you please show the relationships in Fig. 3 using linear

scales? I think that it is necessary, since the scatter is so large even in the log-log plot, that the readers deserve to see the linear scales too. Any of these can go to the appendix, if it works better.

(5) I find it really great that you were able to predict the map of abundance for every single species, it is a really useful product of your study! Would you be able to give that product away to the readers, e.g. to deposit the maps somewhere where they can be freely and easily downloaded, species by species?

(6) Along similar lines, one lines 156-160 and 212-213 you should give very specific examples of these species (two or three of them) directly in the text. Also, it would be nice to have a list of all these species somewhere, e.g. in the appendix. This is your very cool and relevant result, just give it away to people so that they can use it!

(7) How much does the imperfect detection part of the model (starting on line 311) actually improve the model? Trees are probably the easiest organisms to detect. Would your results change dramatically if you don't include this part and assumed that every detection is an actual presence/absence? I can actually see that there are several "building blocks" that can be added, or removed from the model. These are: (I) The part with eqs. 1-6 modelling the abundance using local plot data, (II) the part on lines 312-328, and (III) a potential environmental covariate. I would suggest to try to omit or add parts II and III, calculate the observed vs. predicted values, and AIC (or DIC), just to see if any of these parts actually contribute significantly to the predictive power of the model. It would be a cool methodological result, in my opinion. But it is all really just a suggestion.

OTHER SPECIFIC COMMENTS

Line 14: understanding of the structure

Line 31: It seems that "novel class" is a bit too strong. Your model is nice, but as you already describe on lines 396-411, it's not a completely new class – it's still a spatial hierarchical model assuming Poisson process and with components that have been used elsewhere.

Lines 41-64: There is some redundancy between descriptions in the two sections, it could be eliminated. Also, you state 40,000 and 40,547, but I guess it would be better to just choose one of these.

Line 54: In 1 sentence briefly state what that gap actually is. Be more specific.

Line 64: I suggest that "We fitted the model using the empirical Bayes procedure" would be more accurate.

Line 65: Are seedlings included? And what is the minimum DBH that you refer to – this is critical, since the estimates can differ by even several orders of magnitude, depending on what you consider to be an individual. It should be clearly defined here.

Lines 67-73: It is not very clear to me how these percentages were obtained. You often cite Crowther et al., and their global estimates, but your estimate is for Japan. Could you be more specific about how were these made comparable?

Line 82: These results should be shown. Ideally in an appendix in a scatterplot.

Fig. 2: Please provide a pairplot showing pairwise correlations between these three variables. Either

here, or in an appendix.

Fig. 4: Use base of 10 for the logarithm, it's easier to infer the actual numbers from that. Or, alternatively, use a log-log plot, but with 1, 10, 100, 1000, 10000 labels. And as I've already mentioned above, readers should get the chance to see these plots on linear scale too, somewhere. If not here, than perhaps in the appendix.

Line 100: Data is plural. Thus, data inform, not informs.

Line 101: I would use an even stronger language here: Modelling may well be the ONLY option to get to these numbers.

Figure 4: Provide labels for the x and y axes of the histograms. I am not sure what the numbers mean.

Line 113: Pattern was.

Line 114: Remove "relatively", it feels redundant.

Line 131: "for species" not necessary.

Table 2: Give more specific units for the abundance. E.g. number of individuals with DBH > 10 cm, or whatever is appropriate.

Fig. 5: Please provide some uncertainty bars along the y axis.

Throughout methods: In English, the word "grid" usually refers to the entire set of cells, but it looks like that by "grid" you understand a single cell. There is some confusion here. I suggest that you stick to the usual definition. Unless I am getting something wrong (I am not a native English speaker).

Along eqs. 4-6, one the latest Otso Ovaskainen's paper in Ecology Letters on JSJM comes to my mind, maybe it could be cited.

Line 337: Maybe you can start a new named section here?

Line 367: When you mention priors: Are any of your parameters actually biologically meaningful that there can be an actual informative prior that could be obtained for them? Wouldn't it improve model fit? An idea to do, or a point for discussion, e.g. at this point.

Eqs. 15, 14, 13, and 12: I am not so sure what all of these likelihoods are for. Which one is the most important one that has been used for model fitting? Either simplify the text. e.g. by removing the unnecessary ones, or move them to an appendix, or just try to add an extra 1-2 sentences to each of these likelihoods, stating in a non-statistical language what they are good for. They look scary.

Line 378: Maybe you can start a new named section here?

Line 436: I am not that familiar with the empirical Bayes procedure, but doesn't it provide actual credible intervals? Or in other words: What is the advantage of having standard errors?

Line 440-441: It is unclear to me where in the analysis was the "area of natural forest in each grid" actually used, but I might have missed it somewhere. Or are these the a_{jk} values? Maybe readers

could be reminded here what was it used for.

Lines 449-460: I am a bit suspicious about the varying DBH across the different datasets here. How was this standardized for the purpose of comparison in Figure 3? And are you sure that the observed tendency of your model to under-predict can't be just caused by some mismatches in DBH? Just something I'd double check.

Lines 442-467: Please mention, for each of this dataset, what is its availability (e.g. if it is open or not).

Lines 461-464: It would be good to know a bit more detail on when and how the data were actually collected, what was the minimum DBH, if the data were used in other study, and so on. 2-3 sentences should be sufficient.

Line 603 – Data Availability. It really is not clear what “reasonable request” means. It would be much better to be transparent and simply state the exact conditions under which the data will be provided.

Good luck!

Reviewer #3:

Remarks to the Author:

Please see attachment.

Review of: “Integrating multiple sources of ecological data to
unveil macroscale species abundance”
(manuscript number NCOMMS-19-01868)

March 7, 2019

Summary of Review

This manuscript describes an analysis of data which informs floristic biodiversity patterns in the East Asian Islands (primarily Japan). It is a well presented paper, that is easy to read, and the data appears to be a great resource and inference from it deserves to be well read and appreciated.

The analysis contains multiple steps. First the data are modelled on a grid cell level using a hierarchical model. The model contains random terms for species, cells and their interaction (no environmental gradients and no spatial terms). Next point estimates of richness and abundance are estimated, and then these are converted to SADs and other biodiversity metrics. The statistical model is mostly coherent and sensible, although I do wonder what role uncertainty in the estimates plays in the final inference.

The manuscript then shows how selected patterns of biodiversity change over the Islands and provides some discussion about why these patterns may be observed.

In general, I like the manuscript. However, I am concerned about some technical aspects of the work. Primarily, this concern revolves around how this method compares with other competing methods (the prediction performance of this method is not overly convincing), and details of the model formulation – I wonder if the inferences could be made to be better. These concerns are described in more detail below in the general and specific comments. The general comments relate to the overall methods and results while the specific comments relate to specific locations in the manuscript. Sometimes specific does not equate to ‘minor’, but most often it does.

General Comments

Uncertainty The authors go to some effort to specify a sensible model, which is then estimated in a statistically sound manner. By this I mean that estimates of parameter uncertainty should be readily available. However, in the summary of the model (into SADs etc) this uncertainty appears to be ignored. How does uncertainty affect the inferences you present? I realise that this is not a trivial task, as tracking uncertainty from the estimates through the next steps could be challenging. However, with MCMC samples it is easily done, and with maximum likelihood estimates one could always take samples of the parameters from their (joint) asymptotic distribution.

Lack of comparison Care needs to be taken when presenting a statistical methodology as a new approach. For this to be taken seriously, the new method needs to be quantitatively compared against

competing methods over many data sets or simulations. Also comparisons should be made on a more theoretical ground. However, that does not mean that the method is wrong or bad. It just means that, in my opinion, you should tone down the claims that this is a new method and just present it as a ‘sensible analysis of a data set’. I realise that you do mention other methods (late in the methods section), but the theoretical comparison is fleeting and sometimes is little more than a pointer to literature.

Utility in adding auxillary data The data set contained 4684 grid cells, and 3695 of these contained at least one plot. This is a fairly high proportion (almost 80%). This makes me wonder about how much extra information is obtained from adding the auxiliary data? Does it change the point predictions? Does it decrease the uncertainty in the inferences? From a methodological perspective, this would be nice to know.

Model fit I am really pleased to see that you included external data sources to test your model. However, the model doesn’t seem to fit these data sources very well (correlation statistics as low as 0.28). No discussion about why these poor fits occur. Are there differences in collection methods? Geographical ranges? Is this to do with shrinkage (although that might make things better)? The reader is left to wonder if they should believe the results.

Independence of data sources Something that concerns me, and perhaps I am alone, is the independence of data sources. For example, what information are the range maps based upon? Does it (formally or informally) incorporate the plot data, or the species occurrence records? Is the plot data incorporated into the occurrence records? Further, these data sources are currently being treated equally – should they be? Should the range maps be treated like a prior, rather than a data source?

Specific Comments

Abstract I tend to disagree with a number of the statements made in the abstract. Notably, there is some literature (not terribly well cited but in relatively good journals) that concerns prediction of RADs¹ over large spatial scales. This approach requires modest amounts of data. Have a look at Foster & Dunstan (2010); Dunstan & Foster (2011); McCarthy *et al.* (2018).

Line 12 ‘...themselves of ecologically relevant as ...’ → ‘...themselves of ecological relevance as...’

Line 20 A reference is needed after this statement.

Line 27 This sounds odd to me. SADs do not account for species labels, and so how can they generate species for extinction risk? Sure, they can inform which communities may have species that are at risk, but not the actual species themselves?

Line 64 Please append ‘as defined in the methods’.

Line 78 If latitude is important (and one suspects that it is likely, along with altitude), then why not formally include it in the model and estimate (with uncertainty) the effect?

Lines 81-84 Without more detail, this sentence is almost meaningless. Please revise or remove.

¹a RAD is similar to a SAD but has a slightly different formulation, see McGill *et al.* (2006)

Figure 2 How are the marginal plots created? Is the line an average? Is it an integration of some sort?

Figure 3 For the FDP data, there seems to be both over and under prediction. Is this shrinkage? It seems fairly severe (for NFI too).

Figure 4 The plot are missing axes labels. Is it obvious what they should be? To a more casual reader?

Line ~160 There seem to be some VU and NT species that could be CR. At least they appear in the bottom left hand corner of the plot. Is this worth mentioning?

Line 163 Are you estimates accurate? I have found no evidence to suggest that they are.

Line 168 Where is this shown? There has been no comparison with and without the auxiliary data.

Line 221 There are many reasons that the model may be under-predicting. Here are some ideas off the top of my head: species expert distribution being too narrow; simple model is too simple and smooths out some of the signal, and; missing covariates could increase responsiveness. The Poisson assumption is not necessarily the most likely cause (and may be one of the harder problems to fix).

Equations 3, 4, 5 & 6 Is it appropriate that these are all i.i.d.? Species covariance? Spatial patterns? Perhaps you want to state what the assumption is and that it is made for convenience rather than ecological plausibility.

Lines 292-297 I wouldn't have thought that adding covariates into the model would have been onerous. Inter-specific covariances are more problematic though. Not including them is making a strong assumption though, which needs to be highlighted.

Line ~310 I think that there is some literature missing here on moving from binary data to abundance data. It seems to me that a lot is dependent on the assumptions made. How wrong can you get it just by making different plausible assumption.

Line 325 What is an individual here?

Equations 8, 9, 10 & 11 why is there no interaction term here? It seems odd to leave it out.

Line 340 What is x_{ij} ? It has not been defined. On first reading I thought that it was a typo and should have been z_{ij} , but I now think not (but don't know what it is).

Line 344 I don't think that you mean 'marginalised' here? What is this marginal to? $z_{ij} = 1$ is conditional.

Line 395 Currently the 'Related Models' section is not very informative in terms of how your model relates. Also the classes of models mentioned seem to be pretty disparate.

Line 515

References

- Dunstan, Piers K., & Foster, Scott D. 2011. RAD biodiversity: prediction of rank abundance distributions from deep water benthic assemblages. *Ecography*, **34**(5), 798–806.
- Foster, Scott D., & Dunstan, Piers K. 2010. The Analysis of Biodiversity Using Rank Abundance Distributions. *Biometrics*, **66**(1), 186–195.

- McCarthy, James K., Mokany, Karel, Ferrier, Simon, & Dwyer, John M. 2018. Predicting community rank-abundance distributions under current and future climates. *Ecography*, **41**(9), 1572–1582.
- McGill, Brian J., Enquist, Brian J., Weiher, Evan, & Westoby, Mark. 2006. Rebuilding community ecology from functional traits. *Trends in Ecology & Evolution*, **21**(4), 178 – 185.

The authors use spatially replicated multispecies detection-non-detection data, along with supplementary information on species occurrences in the Japanese archipelago, to infer geographic woody plant species abundances in natural forests at the 10 x 10 km scale of resolution. Multi-species distribution distributions (SADs) for trees are constructed from the data. The constructed SADs are then used to determine the most likely sort of macroevolutionary speciation that has occurred, which showed that protracted speciation (versus point mutation and fission) is the most likely process. The authors were also able to estimate speciation rates in four biogeographical region. In addition, the authors show from the SADs that it is possible that some woody plant species not currently on the Japanese red list may be vulnerable to extinction.

The authors' analysis addresses a key problem raised by the unified neutral theory of biodiversity and biogeography (UNTB), that the SAD is sensitive to the mode of speciation (e.g., Rosindell et al. 2010, Etienne and Haegeman 2011). The current paper answers the need, expressed in those papers, for getting beyond local communities to macroscale metacommunity data sets to adequately address the question of what mode of speciation is best, and clearly protracted speciation is able to fit the data very well (Figure 4) and predict reasonable values of metacommunity size, fundamental biodiversity number, speciation rate, and average species lifetimes for different biogeographic regions (Table 2). This seems to be a nice contribution to biodiversity theory.

The use of the authors' SADs to generate data on species at the range of low abundances and small areas of occupancy, is a useful step towards determining if the species in this range that are not currently classified, might need to be listed if further conditions (e.g., species are declining in abundance or habitat fragmentation is occurring) are met.

I have a few comments.

Thank you very much for your thoughtful and detailed review. We revised the manuscript according to your comments; please see our point-by-point responses below.

[1-A] Table 2. The units of speciation rate, ν , need to be given. These are 'probability of speciation event per birth in the metacommunity' (page 114, Hubbell 2001).

Thank you for your suggestion. In the revised manuscript, Table 2 has been renumbered as Table 3. Instead of adopting the Hubbell's description, we used a simpler description for ν as "speciation rate (per individual per generation)" in terms of that νJ_M gives the (expected) number of new species per generation (Ricklefs 2003) (caption of Table 3).

[1-B] The methodology of estimating macroscale SADs from spatially replicated detection-nondetection observations using a hierarchical model to link probability of species detection within a plot to density of species in grid cell, described on pages 14-21, is well founded on Royle and Dorazio (2008), so I believe it is basically correct and is easier than doing counts. However, I am wondering about possible underestimation of abundance of the species that are superabundant. If only detection is used to infer abundance, it seems possible that even a large number of replicate plots may not be able to discern the difference between a species that is superabundant on 50% of plots and one that occurs in much lower numbers on the same 50% of plots. Figure 3 seems to indicate that species with higher abundance are more likely to have their abundance underpredicted by the model. For particular cases of underprediction this could be checked. (The authors suggest another reason for underprediction based on assumption of homogeneous Poisson spatial distribution, which also seems reasonable.)

In order to examine the details of predictive performance of the model, we conducted a species-base evaluation of RMSE and bias of prediction which was described in Appendix S2. Although underprediction was not the case for abundant species all the time, the result suggests that some abundant species were indeed considerably underpredicted (Fig. S4). Although the tendency of underprediction was rather caused by an extra-Poisson variation (Appendix S2, lines 139-155), superabundant species could be underpredicted even when the Poisson assumption holds; this may occur because, under the specified observation model, the detection probability saturates as the increase of individual density (discussed in lines 386-392).

[1-C] I would like to know more of the details of the 40,000 vegetation surveys. Was the tree size (> 5 cm DBH) used for determining occupancy, as in the FDP data used for validation, or were smaller trees also included? Also, if forest with closed canopy was predominately sampled, early successional species could be missed.

We detailed more about the vegetation surveys (lines 471-476 and 485-494). The objective of vegetation surveys was to describe the species composition of plants within a landscape rather than to study structure and dynamics of climax forest, and therefore sampling plots were not placed selectively to forest with closed canopy (indeed, we had another ~30k plots that were collected outside of natural forests which we excluded from the analysis).

[1-D] Page 22. The plot sizes range from 0.01 m² to 18,000 m² (page 22), which is an enormous difference. I think there is a typo and the authors meant 0.01 ha. Even so, the difference is large. Is this likely to create any sampling biases?

The range of plot size was correct and even possible for vegetation surveys in which the area of a plot can vary depending on the vegetation type, specifically plant canopy height. For example, a plot can be significantly smaller when the vegetation is homogenous and includes only small bushes (0.01 m² may be somewhat an extreme case, though). We don't expect that the variation in plot area induces a bias. Given that detection probability depends on the product of individuals density of species and plot area (Equation 7), it would rather help to infer different levels of individual density.

[1-E] Also, it would be good to know what time period the surveys were done. It is not likely that species compositions changed too much over the period of the surveys, but it is useful to know the time period in any case. Perhaps more importantly, species occurrence data were also used from other sources, such as museum and herbarium specimens and early distribution maps (Lines 429-433). Inclusion of these old records carries some risk.

We described details of the time period of the vegetation surveys (lines 474-476 and Supplementary Fig. S1C). We acknowledge that the inclusion of old records is a limitation of our study and discussed some temporal aspects of the inference in Discussion (lines 290-297).

[1-F] False positives could also occur through misidentification. Is there some quality control on the data that attempt to avoid that?

We scrutinised taxonomic uncertainty, including possible species misidentification, of each species. Specifically, we first checked all taxa by reviewing their standard scientific names following the woody plant nomenclature used in the Japanese-Scientific Names Index (Yonekura & Kajita, 2003), and then pooled data of subspecies and varieties into that of their parent species (lines 480-482). This treatment should filter out false positives of related species that could be misidentified.

K. Yonekura and T. Kajita (2003) BG Plants: index for Japanese scientific plant names: Ylist.
<http://ylist.info/>

[1-G] Are only native woody plants included, versus invasive woody plants that have become established?

Both native and introduced species are included in the analyses, except for the analysis in “Implication for biodiversity conservation”. We added an explanation to the main text (lines 55-57) and in the Methods section (line 483).

[1-H] Validation is well described on Page 23, and Figure 3 gives a useful summary, but it would be interesting to know if there were particular species that were very poorly estimated, and perhaps why.

We are grateful for your suggestion. We examined the predictive performance of the model for each species. The result of this new analysis indicated that: (1) Although the estimation bias was relatively small for most species, some species with greater individual density and/or narrower geographic range indicated a particularly substantial negative bias, i.e. under-estimation; (2) The predictive errors were indeed positive in most cases, implying that a major factor of the negative bias was ascribed to the prediction failure of extremely high species abundance occurring with lower frequency, i.e. aggregated (patchy) distribution of species. The greater negative bias observed in species with narrower geographic range is likely caused by the limited precision of the inference for them due to a limited number of grid cells that they are present. This analysis is now described in Appendix S2 (lines 139-155) and mentioned in the main text (lines 76-78 and 209-217). Although not described in the manuscript, we also tried to test some relationship between the goodness of prediction and functional traits of species (e.g. their maximum height and the mode of dispersal). In this analysis, however, we were not successful to find a reasonable relationship. Please also see our response to the comment 1-B.

Minor comments:

Line 60. Change ‘comprised of’ to ‘comprising’

Line 64. Change ‘abundance’ to ‘abundances’

Line 65. Change ‘estimated to’ to ‘estimated to be’

Line 81. I am not sure what ‘a more refined pattern’ means here.

Figure 3 caption. Change ‘crossed lines’ to ‘diagonal lines’

Line 141. Change delete ‘are’

Line 314. Insert comma after ‘inaccurate’

Line 340. Change ‘denotes’ to ‘denote’

Line 515. Change ‘proximate’ to ‘approximate’ and insert comma after ‘definition’

Line 540. Change ‘for these’ to ‘for each of these’

We corrected as pointed. For Line 81 (line 63 in the revised manuscript), we changed ‘a more refined pattern’ to ‘a less-autocorrelated pattern.’

Reviewer #2 -----

The present manuscript presents a novel approach to tackle several fundamental issues of spatial ecology: Where species live, how abundant they are, and can we derive particularly useful conclusions from their distributions and abundance. Not only does the work address all of these simultaneously, it also offers a clear methodological advancement by integrating everything in a single and elegant statistical model. To add further novelty, the analysis/model also explicitly operates at several spatial scales. Finally, the dataset (Japanese trees) is large and solid, and adds robustness and credibility.

Apart from the clear methodological and conceptual advancement, there are some highly relevant and striking results. The first useful result is the suite of maps of abundance for each species (they should definitely be provided with the paper). On top of that, the authors infer, from the estimated large-scale SAD, an elegant and straightforward support for the protracted mode of speciation, and they predict that there are species whose conservation status should be revisited. Thus, their approach and results are relevant for both basic macroecology and macroevolutions, as well as for applied conservation.

Although this seems like a lot to be offered by a single study, it all fits together naturally, mostly because of everything actually follows from a single statistical model. The manuscript is also very well written, I rarely get the chance to review something that well crafted. Finally, I applaud the clarity of the methods description and the clear mathematical notation. Overall, I think that this can become a milestone and highly cited reference in the research of large scale biodiversity patterns and data integration. As such it is potentially suitable for Nature Communications.

I have, however, some reservations that prevent me from recommending the ms for publication in its current form.

Thank you very much for your detailed and constructive comments. We implemented most of your suggestions; please see our point-by-point responses below.

MAJOR COMMENTS

[2-A] (1) I am really surprised that the authors haven't considered a single environmental predictor in their model. It is really well established that tree distributions, species richness, biomass, or densities can be predicted by variables such as NPP, AET, biomass, NDVI, or some combination of precipitation- and temperature-related climatic variables. Further, these variables are really easy to get at some good resolutions such as 1 x 1 km or even 250 x 250 m – extracting them for the plot or grid data is a matter of one day with raster package in R and function “extract”. On lines 289-297 the authors give some justification for why they haven't included this, but the arguments are not very convincing. The environmental predictors don't have to have a specific parameter for each species – they can operate at a higher hierarchical level, e.g. affecting the total density of all species, thus increasing the total number of parameters by just a couple. Moreover, it can even be that if the right predictor is included, both the parameter identifiability and predictive power can improve! I've seen MCMC algorithms in Stan that converged faster (i.e. a single MCMC iteration was actually faster) for more complex models because the models were simply more reasonable. For a journal such as Nature Communications I would expect the study not to have such an obvious omission. Thus, I strongly suggest that the authors try to add at least 1 or 2 environmental (climate-related or productivity-related) predictors to some part of the model (e.g. predicting d_{ij}). If it is too computationally demanding to use the predictor on a per-species basis, then its effect could be modeled on some higher level, e.g. as a predictor of the total N of all individuals of all species in a cell (or plot, or both). I am aware that this is extra work, potentially protracting the publication process, but it would make a difference between a regular specialist macroecological analysis, and a top-level Nature Communications study. And the current predictive power of the model really asks for improvement, and the role of climate and productivity is just too notorious to be ignored. I am also aware that there may be some problems with this idea, and that you will possibly find out that the model is still better off without such predictors, but I'd like you to at least try.

Thank you very much for constructive comments. We totally agree with the importance of environmental predictors. In this project, we firstly have tried to account for species-specific responses to environmental factors, but finally regarded it as a future task because of the current

computational limitation. We thought at the same time that exclusion of covariates likely allows us macroecological post-hoc analyses in which effects of environmental variables on estimated biodiversity patterns could be examined in a non-tautological manner. However, the comment led us to realise correctly that “the environmental predictors don’t have to have a specific parameter for each species,” and that such a treatment would decrease the risk of an inappropriate description of the patterns of biodiversity caused by the whole omission of covariates. According to this suggestion, we included environmental covariates in the model and examined a series of different models with/without covariates. Please see Appendix S2 for details of the model comparison. To identify which kind of environmental factors should be involved to improve the inference, we examined the correlation between various environmental covariates that we could obtain (including climatic, topographic, and pedochemical factors) and grid-cell random effects predicted in a model without covariates. As a result, we selected actual evapotranspiration (AET) as a variable related to climate and the human influence index (HII) as a variable related to human impact. Thanks to your practical advice, the inclusion of the covariates successfully improved model prediction. In the revised manuscript, we specifically reported the result of the model with the two covariates (in addition to the interaction term) as the main result.

[2-B] (2) Along similar lines, I miss at least a mention of the role of human impact on the Japanese forests, and on the parameter estimates and predictions. I would expect this to be really important in a country with high human population density such as Japan. Ideally, this could also be incorporated to the model as a predictor, but I am aware that this could really be too much (although the GIS layers are there). At least it should be discussed.

The potential human impact on the geographic patterns of species abundance is now accounted for by the inclusion of the HII covariate that is an inclusive measure of human activities (lines 508-510; please also see Appendix S2 and Figs. S2-S3). The analysis suggested that human activities are likely to influence the species abundance of woody plants (lines 64-70, 179-194).

[2-C] (3) Would spatial autocorrelation be hard to incorporate? And which part of the model would be the best to incorporate it? Could actually improve the predictions? Again, I see this as optional, but it should at least be mentioned.

Including spatial autocorrelation in the model was rather difficult in our application. We have tried to introduce spatial correlation in the two cell-level random effects ($e^{(2)}_j$ and $u^{(2)}_j$), but the inference was computationally fairly expensive even with the simplest (parsimonious) model. Given a limited computational resource, we were then only able to compare models without spatial random effects. We mentioned this point in Appendix S2 (lines 74-76) where we described the preliminary model comparison in detail.

[2-D] (4) The model validation (Fig. 3) shows that the predictive power at local scale is not that great. It may not be such a big problem since the main inference is done on the regional scale. Anyway, I can see that an additional validation on a larger scale can be made by comparing the total predicted abundance at the 10 x 10 km cells with estimates from other studies, e.g. from Crowther et al. This should be an easy extra plot to add – you would just aggregate the values from Crowther over the 10 x 10 km grid cells. In addition, it would be good to have the relationship mentioned on line 82 shown graphically, in a scatterplot. Finally, would you please show the relationships in Fig. 3 using linear scales? I think that it is necessary, since the scatter is so large even in the log-log plot, that the readers deserve to see the linear scales too. Any of these can go to the appendix, if it works better.

We obtained estimates of tree abundance for 10 km grid squares that are based on Crowther et al. (2015) and compared them with the results of our model. Results are reported in lines 79-87. As we

mentioned initially (number 10 of the major points of the revision), the comparison between the estimates of the model and that of Kubota et al. (2015) were removed from the current manuscript. The scatter plots in linear scales are provided in Supplementary Fig. S6.

[2-E] (5) I find it really great that you were able to predict the map of abundance for every single species, it is a really useful product of your study! Would you be able to give that product away to the readers, e.g. to deposit the maps somewhere where they can be freely and easily downloaded, species by species?

Thank you very much. We provided the abundance map of all species (but except for threatened species) in Appendix S4.

[2-F] (6) Along similar lines, one lines 156-160 and 212-213 you should give very specific examples of these species (two or three of them) directly in the text. Also, it would be nice to have a list of all these species somewhere, e.g. in the appendix. This is your very cool and relevant result, just give it away to people so that they can use it!

We gave some names of species in the main text (lines 274-276) and added a list of native species potentially threatened in Appendix S4.

[2-G] (7) How much does the imperfect detection part of the model (starting on line 311) actually improve the model? Trees are probably the easiest organisms to detect. Would your results change dramatically if you don't include this part and assumed that every detection is an actual presence/absence? I can actually see that there are several "building blocks" that can be added, or removed from the model. These are: (I) The part with eqs. 1-6 modelling the abundance using local plot data, (II) the part on lines 312-328, and (III) a potential environmental covariate. I would suggest to try to omit or add parts II and III, calculate the observed vs. predicted values, and AIC (or DIC), just to see if any of these parts actually contribute significantly to the predictive power of the model. It would be a cool methodological result, in my opinion. But it is all really just a suggestion.

We compared these potential modelling options in which we confirmed that the inclusion of data integration and covariates improved inference. Please see Appendix S2.

OTHER SPECIFIC COMMENTS

[2-a] Line 14: understanding of the structure

We corrected as pointed (now in line 6).

[2-b] Line 31: It seems that "novel class" is a bit too strong. Your model is nice, but as you already describe on lines 396-411, it's not a completely new class – it's still a spatial hierarchical model assuming Poisson process and with components that have been used elsewhere.

We improved this statement not to over-emphasize the novelty of the model as follows (lines 26-28):

In this view, we developed a hierarchical modelling framework (Royle & Dorazio 2008, Kéry & Schaub 2012, Kéry & Royle 2016) that estimates species abundance over a large geographic extent, which we named "macroscale species abundance".

[2-c] Lines 41-64: There is some redundancy between descriptions in the two sections, it could be eliminated. Also, you state 40,000 and 40,547, but I guess it would be better to just choose one of these.

We deleted the first paragraph of the section *An application to woody plant communities in East Asian islands*. We also rephrased “more than 40,000” to state the exact number of plots (line 38).

[2-d] Line 54: In 1 sentence briefly state what that gap actually is. Be more specific.

We stated more concretely as follows (lines 50-51):

i.e. some species with limited abundance and area of occupancy may have been underrated in a national red list.

[2-e] Line 64: I suggest that “We fitted the model using the empirical Bayes procedure” would be more accurate.

This sentence has been deleted.

[2-f] Line 65: Are seedlings included? And what is the minimum DBH that you refer to – this is critical, since the estimates can differ by even several orders of magnitude, depending on what you consider to be an individual. It should be clearly defined here.

We concretely described what kind of individuals are included as follows (lines 55-57):

The total abundance of woody plants (which encompasses introduced species and nominally includes individuals of any size; see Estimating macroscale species abundance of woody plant communities in East Asian islands in the Methods section) revealed ...

[2-g] Lines 67-73: It is not very clear to me how these percentages were obtained. You often cite Crowther et al., and their global estimates, but your estimate is for Japan. Could you be more specific about how were these made comparable?

Unlike the previous version of the manuscript in which we used the global estimate of Crowther et al., in this revision we obtained their estimates at the 10km grid scale that can be compared directly with our estimate (we are grateful to the reviewer providing the comment 2-D that led us to get this additional dataset). Please see lines 551-568 for details.

[2-h] Line 82: These results should be shown. Ideally in an appendix in a scatterplot.

As we noted above, the comparison between the estimates of the model and that of Kubota et al. (2015) were removed from the current manuscript.

[2-i] Fig. 2: Please provide a pairplot showing pairwise correlations between these three variables. Either here, or in an appendix.

We added a scatter plot matrix in Supplementary Fig. S5.

[2-j] Fig. 4: Use base of 10 for the logarithm, it's easier to infer the actual numbers from that. Or, alternatively, use a log-log plot, but with 1, 10, 100, 1000, 10000 labels. And as I've already mentioned above, readers should get the chance to see these plots on linear scale too, somewhere. If not here, than perhaps in the appendix.

We understand this comment is about Fig. 3 rather than 4. The figure was shown in the logarithms of base 10, but the label was confusing. We revised accordingly. As mentioned, the figure of linear scales is given in Supplementary Fig. S6.

[2-k] Line 100: Data is plural. Thus, data inform, not informs.

Thanks, but this part has been deleted as a result of our response 2-l (please see below).

[2-l] Line 101: I would use an even stronger language here: Modelling may well be the ONLY option to get to these numbers.

We revised to be more strongly worded as follows (lines 100-102):

As a solution to this problem, the estimates of macroscale species abundance may be useful, and probably is the only option given that obtaining data on species abundance over a huge spatial extent is obviously unrealistic.

[2-m] Figure 4: Provide labels for the x and y axes of the histograms. I am not sure what the numbers mean.

We corrected as pointed.

[2-n] Line 113: Pattern was.

We corrected as pointed (line 113).

[2-o] Line 114: Remove “relatively”, it feels redundant.

We corrected as pointed (line 115).

[2-p] Line 131: “for species” not necessary.

We corrected as pointed (line 132).

[2-q] Table 2: Give more specific units for the abundance. E.g. number of individuals with DBH > 10 cm, or whatever is appropriate.

We described what the abundance means. Please note that the previous table 2 is now numbered as Table 3.

[2-r] Fig. 5: Please provide some uncertainty bars along the y axis.

We understand the importance of assessing estimation uncertainty. Nevertheless, it was technically very challenging to quantify the uncertainty of the estimates of species abundance and that of the area of occupancy of species because of their high dimensionality. Actually, TMB package, which we have used to fit the model, provides the functionality to calculate asymptotic standard errors for unknown quantities that are a function of random effects and parameters. Although we have tried to use it to estimate uncertainty, we found that the calculation does not finish even after a number of days of running. Given that matrix inversion (which is required to obtain asymptotic standard errors) has a computational complexity $O(N^3)$, it appears that this approach is not practical to our specific application in which millions of random effects (i.e. 4k grid cells by 1k species) are

involved. We are aware of that uncertainty can be well quantified by using the alternative fully Bayesian approach, but this option was also impractical for us. We have tried Stan, a fast gradient-based MCMC sampler, but it showed no sign of completing the required posterior sampling within a reasonable period of running (indeed, this was why we used TMB, the empirical Bayes approach, to fit models). Thus, in this study, we were inevitable to avoid doing the uncertainty assessment for random effect parameters and quantities obtained as a function of them. We added a new paragraph in Discussion mentioning that this was a major limitation of our inference (lines 284-289).

[2-s] Throughout methods: In English, the word “grid” usually refers to the entire set of cells, but it looks like that by “grid” you understand a single cell. There is some confusion here. I suggest that you stick to the usual definition. Unless I am getting something wrong (I am not a native English speaker).

We replaced “grid” with “cell” or “grid cell” where appropriate.

[2-t] Along eqs. 4-6, one the latest Otso Ovaskainen’s paper in Ecology Letters on JSDM comes to my mind, maybe it could be cited.

In response to a comment from the reviewer 3 (comment 3-v), we decided to remove the “Related Models” section.

[2-u] Line 337: Maybe you can start a new named section here?

We broke the section and named the new section as “Likelihood of the model” which were now moved to Appendix S1.

[2-v] Line 367: When you mention priors: Are any of your parameters actually biologically meaningful that there can be an actual informative prior that could be obtained for them? Wouldn’t it improve model fit? An idea to do, or a point for discussion, e.g. at this point.

We decided not to mention the possibility of informative priors. We understand that proactive use of biologically informative priors is a contentious issue on which there is no consensus yet, but it is not recommended generally to achieve an objective inference (Kéry and Royle 2016, section 2.5.3). Our basic stance is that available information should be rather utilised in a form of integrating all data within a common statistical model, but not in a form of an informative prior.

[2-w] Eqs. 15, 14, 13, and 12: I am not so sure what all of these likelihoods are for. Which one is the most important one that has been used for model fitting? Either simplify the text. e.g. by removing the unnecessary ones, or move them to an appendix, or just try to add an extra 1-2 sentences to each of these likelihoods, stating in a non-statistical language what they are good for. They look scary.

We moved this part to Appendix S1.

[2-x] Line 378: Maybe you can start a new named section here?

We broke the section and named the new section as “Ecological variables obtained as derived quantities” (line 426).

[2-y] Line 436: I am not that familiar with the empirical Bayes procedure, but doesn’t it provide actual credible intervals? Or in other words: What is the advantage of having standard errors?

Please see our response to the comment 2-r. The delta method can be used to obtain asymptotic standard errors in the empirical Bayes procedure (Fournier et al. 2012). Nevertheless, this was rather impractical for our application.

D. A. Fournier , H. J. Skaug , J. Ancheta , J. Ianelli , A. Magnusson , M. N. Maunder , A. Nielsen and J. Sibert (2012) AD Model Builder: using automatic differentiation for statistical inference of highly parameterized complex nonlinear models. *Optimization Methods and Software* 27:233-249.

[2-z] Line 440-441: It is unclear to me where in the analysis was the “area of natural forest in each grid” actually used, but I might have missed it somewhere. Or are these the a_{jk} values? Maybe readers could be reminded here what was it used for.

We defined A_j , the area of natural forest in cell j , in an early part of the model description (lines 334-335).

[2-a'] Lines 449-460: I am a bit suspicious about the varying DBH across the different datasets here. How was this standardized for the purpose of comparison in Figure 3? And are you sure that the observed tendency of your model to under-predict can't be just caused by some mismatches in DBH? Just something I'd double check.

We discussed the effect of the difference in the size of individuals between the fitted model and validation datasets to the predictive ability of the model (lines 195-208). The difference of DBH between datasets (including GMFT, the new dataset) was not standardised. It allowed us, however, a detailed examination of the prediction bias in which the observed pattern of bias appeared to be well explained by the difference of DBH between datasets.

[2-b'] Lines 442-467: Please mention, for each of this dataset, what is its availability (e.g. if it is open or not).

We mentioned explicitly that validation datasets, except FSLE, are publicly available (lines 528-530, 536-537, and 554-556). The FSLE dataset is mentioned as unpublished data, so it is not open by definition.

[2-c'] Lines 461-464: It would be good to know a bit more detail on when and how the data were actually collected, what was the minimum DBH, if the data were used in other study, and so on. 2-3 sentences should be sufficient.

We added some details about how the FSLE data were collected (lines 538-542).

[2-d'] Line 603 – Data Availability. It really is not clear what “reasonable request” means. It would be much better to be transparent and simply state the exact conditions under which the data will be provided.

We revised the Data Availability statement. The datasets have been registered in GIVD and are available under the GIVD policies.

Reviewer #3 -----
Summary of Review

This manuscript describes an analysis of data which informs floristic biodiversity patterns in the East Asian Islands (primarily Japan). It is a well presented paper, that is easy to read, and the data appears to be a great resource and inference from it deserves to be well read and appreciated.

The analysis contains multiple steps. First the data are modelled on a grid cell level using a hierarchical model. The model contains random terms for species, cells and their interaction (no environmental gradients and no spatial terms). Next point estimates of richness and abundance are estimated, and then these are converted to SADs and other biodiversity metrics. The statistical model is mostly coherent and sensible, although I do wonder what role uncertainty in the estimates plays in the final inference.

The manuscript then shows how selected patterns of biodiversity change over the Islands and provides some discussion about why these patterns may be observed.

In general, I like the manuscript. However, I am concerned about some technical aspects of the work. Primarily, this concern revolves around how this method compares with other competing methods (the prediction performance of this method is not overly convincing), and details of the model formulation – I wonder if the inferences could be made to be better. These concerns are described in more detail below in the general and specific comments. The general comments relate to the overall methods and results while the specific comments relate to specific locations in the manuscript. Sometimes specific does equate to ‘minor’, but most often it does.

Thank you very much for your careful review and constructive comments. We revised the manuscript in accord with your comments. One thing we were not able to accommodate was the uncertainty assessment, which is computationally very challenging. For details, please see our point-by-point responses below.

General Comments

[3-A] Uncertainty: The authors go to some effort to specify a sensible model, which is then estimated in a statistically sound manner. By this I mean that estimates of parameter uncertainty should be readily available. However, in the summary of the model (into SADs etc) this uncertainty appears to be ignored. How does uncertainty affect the inferences you present? I realise that this is not a trivial task, as tracking uncertainty from the estimates through the next steps could be challenging. However, with MCMC samples it is easily done, and with maximum likelihood estimates one could always take samples of the parameters from their (joint) asymptotic distribution.

Please see our response to the comment 2-r and 2-y in which the reviewer 2 also pointed about the assessment of uncertainty. As you mentioned, there are formal procedures to estimate uncertainty in estimated quantities, both in full Bayes and empirical Bayes approaches. Nevertheless, this was rather challenging for our specific application where we have millions of unknown quantities. For fixed effect parameters, we were able to evaluate asymptotic standard errors which are reported in Appendix S2 (lines 134-137). For random effect parameters (and quantities obtained as a function of them), however, it required a great amount of computation which seems impractical.

[3-B] Lack of comparison: Care needs to be taken when presenting a statistical methodology as a new approach. For this to be taken seriously, the new method needs to be quantitatively compared against competing methods over many data sets or simulations. Also comparisons should be made on a more theoretical ground. However, that does not mean that the method is wrong or bad. It just means that, in my opinion, you should tone down the claims that this is a new method and just present it as a ‘sensible analysis of a data set’. I realise that you do mention other methods (late in

the methods section), but the theoretical comparison is fleeting and sometimes is little more than a pointer to literature.

We revised to avoid saying the model as a “novel class” and toned down the related statements in general not to emphasize novelty too much. In addition, we have deleted the “Related Models” section.

[3-C] Utility in adding auxiliary data: The data set contained 4684 grid cells, and 3695 of these contained at least one plot. This is a fairly high proportion (almost 80%). This makes me wonder about how much extra information is obtained from adding the auxiliary data? Does it change the point predictions? Does it decrease the uncertainty in the inferences? From a methodological perspective, this would be nice to know.

We conducted a model comparison in which a model with and without auxiliary data were contrasted. Details are described in Appendix S2. Although in this analysis we were not able to examine how the data integration can reduce uncertainty in the inference, for the reason explained in response 3-A, it highlighted how the point estimates are improved (which was summarised in Table S1).

[3-D] Model fit: I am really pleased to see that you included external data sources to test your model. However, the model doesn't seem to fit these data sources very well (correlation statistics as low as 0.28). No discussion about why these poor fits occur. Are there differences in collection methods? Geographical ranges? Is this to do with shrinkage (although that might make things better)? The reader is left to wonder if they should believe the results

We added a new paragraph to discuss this issue (lines 218-233). Although a number of potential reasons for the weak correlation can be imagined, we yet have only limited clues to identify the major factor for it. That said, the result of preliminary model comparison (Table S1) suggested that a poor explanation of geographic variation in species abundance was at least one of the causations and that we may expect further improvement by including additional components (e.g. further covariates and correlated random effects) to the model to account for variation in species abundance explicitly. Another potential factor was the lack of data that are adequate for the fitted model.

[3-E] Independence of data sources: Something that concerns me, and perhaps I am alone, is the independence of data sources. For example, what information are the range maps based upon? Does it (formally or informally) incorporate the plot data, or the species occurrence records? Is the plot data incorporated into the occurrence records? Further, these data sources are currently being treated equally – should they be? Should the range maps be treated like a prior, rather than a data source?

The two different kinds of data, i.e. plot-level replicated detection-nondetection observations and cell-level presence-absence information, are assumed to be independent. We mentioned this more explicitly (lines 415-416 and 499). As suggested, the auxiliary data sources could also be used in the model as a sort of prior expectation (e.g. Merow et al. 2017). This should be another modelling option right along with other different models for macroscale species abundance that would deserve a formal comparison. Nevertheless, given the limited computational resource (please see Appendix S2), it was beyond the focus of the current study (please also see our response to comment 2-v from reviewer 2 in which we stated our notion about the use of extra information in an inference).

C. Merow, A. M. Wilson and W. Jetz (2017) Integrating occurrence data and expert maps for improved species range predictions. *Global Ecology and Biogeography* **26**:243-258.

Specific Comments

[3-a] Abstract: I tend to disagree with a number of the statements made in the abstract. Notably, there is some literature (not terribly well cited but in relatively good journals) that concerns prediction of RADs over large spatial scales. This approach requires modest amounts of data. Have a look at Foster & Dunstan (2010); Dunstan & Foster (2011); McCarthy et al. (2018).

Thank you very much for pointing out this issue. Your comments (this one, and 3-d below) have corrected our previous conception about SADs. We now realized that the proposed model is not a model for SADs (or RADs) in itself because it retains the identity of species, and that we should, therefore, avoid using “SADs” in most part of the manuscript. We replaced SADs to “species abundance” where appropriate.

[3-b] Line 12 ‘...themselves of ecologically relevant as ...’ → ‘...themselves of ecological relevance as...’

We corrected as pointed (line 5).

[3-c] Line 20 A reference is needed after this statement.

By adding a semicolon, we connected this statement to the following sentence. Our intention was that the appropriate references to the statement are those cited in the next sentence (lines 11-17).

[3-d] Line 27 This sounds odd to me. SADs do not account for species labels, and so how can they generate species for extinction risk? Sure, they can inform which communities may have species that are at risk, but not the actual species themselves?

Please see our response to comment 3-a. We avoided using the term “SADs” and replaced it to “species abundance” to indicate species-specific rarity/commonness (line 19).

[3-e] Line 64 Please append ‘as defined in the methods’.

We deleted this part in response to the comment 2-c from reviewer 2.

[3-f] Line 78 If latitude is important (and one suspects that it is likely, along with altitude), then why not formally include it in the model and estimate (with uncertainty) the effect?

As a response to a comment from reviewer 2 (2-A and 2-B), we improved the model to include two covariates, the actual evapotranspiration (AET) and the human influence index (HII); please see Appendix S2 for details. We regard latitude as a simple descriptor for geographic patterns of biodiversity across the East Asian islands that extend toward NE-SW direction. Given that this geographic gradient is well represented by AET, which was strongly correlated with latitude (correlation coefficient: -0.89, see also Supplementary Fig. S2A) and that AET is directly related to climate, we decided to use AET as a covariate but not latitude.

[3-g] Lines 81-84 Without more detail, this sentence is almost meaningless. Please revise or remove.

As we mentioned initially (number 10 of the major points of the revision), the comparison between the estimates of the model and that of Kubota et al. (2015) were removed from the revised manuscript.

[3-h] Figure 2 How are the marginal plots created? Is the line an average? Is it an integration of some sort?

We added a description of the marginal plots in the figure legend.

[3-i] Figure 3 For the FDP data, there seems to be both over and under prediction. Is this shrinkage? It seems fairly severe (for NFI too).

Shrinkage (or, the lack of enough data) can be related, but other factors might also be relevant as well. We were not able to identify the major factor of the lower correlation in FDP and NFI, although the results suggested that, at least, the model may still lack some components that could further explain variation in species abundance and improve the correlation. Please also see our response to comment 3-D.

[3-j] Figure 4 The plot are missing axes labels. Is it obvious what they should be? To a more casual reader?

We added axes labels.

[3-k] Line ~160 There seem to be some VU and NT species that could be CR. At least they appear in the bottom left hand corner of the plot. Is this worth mentioning?

Thank you very much for your suggestion. We (of course) would not be able to make a definitive statement that VU or NT species in the bottom left hand corner of Fig. 5 can be CR, but we believe that a special attention should be paid to these species. We noted this point in Discussion (lines 272-278).

[3-l] Line 163 Are you estimates accurate? I have found no evidence to suggest that they are.

We deleted “accurate” (line 165).

[3-m] Line 168 Where is this shown? There has been no comparison with and without the auxiliary data.

According to the reviewer’s comments, in the revision, we compared these two kinds of model. Please see our response to comment 3-C.

[3-n] Line 221 There are many reasons that the model may be underpredicting. Here are some ideas off the top of my head: species expert distribution being too narrow; simple model is too simple and smooths out some of the signal, and; missing covariates could increase responsiveness. The Poisson assumption is not necessarily the most likely cause (and may be one of the harder problems to fix).

We conducted some detailed assessments of the predictive performance of the model in which several patterns and potential factors of the underprediction are described and examined. Please see Appendix S2 (lines 139-166) and Discussion (lines 195-217).

[3-o] Equations 3, 4, 5 & 6 Is it appropriate that these are all i.i.d.? Species covariance? Spatial patterns? Perhaps you want to state what the assumption is and that it is made for convenience rather than ecological plausibility.

We explicitly mentioned that the model described in this section has the simplest form to explain the basic modelling idea as follows (lines 319-322):

To describe the basic modelling idea, we here illustrate the models in their simplest form which includes only intercept and independent random effect terms. As described later, however, more complex models could also be developed to incorporate additional ecological plausibility.

[3-p] Lines 292-297 I wouldn't have thought that adding covariates into the model would have been onerous. Inter-specific covariances are more problematic though. Not including them is making a strong assumption though, which needs to be highlighted.

According to the reviewer's comments and suggestions, we added two environmental covariates (AET and HII) to the model, resulting in an improved inference. The covariates were selected based on preliminary analyses of the correlation between various environmental covariates that we could obtain (including climatic, topographic, and pedochemical factors) and grid-cell random effects predicted in a model without covariates. Please see Appendix S2 for details.

[3-q] Line ~310 I think that there is some literature missing here on moving from binary data to abundance data. It seems to me that a lot is dependent on the assumptions made. How wrong can you get it just by making different plausible assumption.

We added a new paragraph to discuss more about the key assumption of a superposed homogeneous Poisson point process before moving to the integrated model (lines 378-392).

[3-r] Line 325 What is an individual here?

It is not different from the "individual" found in the previous part. We think, however, that the introduction of the concept of conditional individual density could be confusing here. Therefore, we added "conditional" and referred related equations to explicitly indicate that our concern is about the modelling of conditional individual density (line 409).

[3-s] Equations 8, 9, 10 & 11 why is there no interaction term here? It seems odd to leave it out.

Such an interaction term can be included formally in equation (9). Nevertheless, given that z_{ij} is unique to species i in cell j (i.e. there is no replication of z for each combination of species and cell), adding an interaction term to this level can be redundant. Actually, we have fitted a model with an interaction term and compared it to a model without interaction. We found that the value of maximum likelihood of this model was the same as the model without interaction, and that the interaction model had an AIC value that was exactly 2 units higher than that of the model without interaction. Therefore, we did not consider this interaction term.

[3-t] Line 340 What is x_{ij} ? It has not been defined. On first reading I thought that it was a typo and should have been z_{ij} , but I now think not (but don't know what it is).

We first note that in the revised manuscript, this part has been moved to Appendix S1 and we use m_{ij} instead of x_{ij} . In the model, z_{ij} , the presence/absence of species i in cell j , is modeled as a variable that is partially observed. m_{ij} is a variable that indicates if the value of z_{ij} is known or not (i.e. their presence or absence is confirmed via the plot-level detection-nondetection observations and/or the auxiliary distribution data). In order to clarify the meaning of this variable, we revised the main text (lines 431-433) in addition to Appendix S1 (lines 33-34).

[3-u] Line 344 I don't think that you mean 'marginalised' here? What is this marginal to? $z_{ij} = 1$ is conditional.

In the revised manuscript, this part has been moved to Appendix S1 (labeled as Equation S1). The latter case is a marginal likelihood in a sense that z_{ij} is marginalised out, i.e. it is expressed as the form: $\Pr(z_{ij} = 1) \times \Pr(\mathbf{y}_{ij} | z_{ij} = 1) + \Pr(z_{ij} = 0) \times \Pr(\mathbf{y}_{ij} | z_{ij} = 0)$.

[3-v] Line 395 Currently the ‘Related Models’ section is not very informative in terms of how your model relates. Also the classes of models mentioned seem to be pretty disparate.

We removed this section.

Reviewers' Comments:

Reviewer #1:

Remarks to the Author:

The authors use spatially replicated multispecies detection-non-detection data, along with supplementary information on species occurrences in the Japanese archipelago, to infer geographic woody plant species abundances in natural forests at the 10 x 10 km scale of resolution. Multi-species distribution distributions (SADs) for trees are constructed from the data. The constructed SADs are then used to determine the most likely sort of macroevolutionary speciation that has occurred, which showed that protracted speciation (versus point mutation and fission) is the most likely process. The manuscript answers the need, expressed in those papers, for getting beyond local communities to macroscale metacommunity data sets to adequately address the question of what mode of speciation is best, and clearly protracted speciation is able to fit the data very well (Figure 4) and predict reasonable values of metacommunity size, fundamental biodiversity number, speciation rate, and average species lifetimes for different biogeographic regions (Table 2). This seems to be a nice contribution to biodiversity theory.

Comments:

I reviewed an earlier version of this manuscript. The authors have responded to the comments and modified or added text in many places.

A minor question. In Figure 4 the left tail is very short, indicating 'negligible' number of rare species (and implying the protracted speciation model). There is evidence from other studies (Magurran and Henderson 2003, Borda-de-Agua et al. 2012) that the number of rare species increases with area, leading to bimodal distribution, in which one of the maxima is for rare species. So there might have been some thought that this trend could continue for larger spatial scales, toward the distribution predicted by the point mutation speciation model. Since this is not observed in the results of this manuscript, it is possibly related to the approach used here, which assumes a homogeneous Poisson distribution?

Table 1, caption, line 2. Change 'dataset' to 'datasets'

Line 64. By 'conditional individual density' is 'conditional on presence' meant?

Lines 67, 68. Change 'whereas' to 'whereas it'

Line 86. By 'billion' is the 'short scale' 10^9 , or the 'long scale' 10^{12} meant?

Line 429. Change 'Equations 3 and 9, respectively' to 'Equations 9 and 3, respectively'

Line 509. Insert comma after 'effects'

This work is a strong contribution to the estimation of species abundance distributions. In particular the use of the SADs to generate data on species at the range of low abundances and small areas of occupancy is a useful step towards determining if the species in this range that are not currently classified, might need to be listed if further conditions (e.g., species are declining in abundance or habitat fragmentation is occurring) are met.

Reviewer #2:

Remarks to the Author:

I really liked the first version of the manuscript, and I am impressed by how carefully and rigorously the authors have addressed the extensive comments from the first round of reviews. My main concern was the lack of environmental predictors in the models, and I see this to be fully addressed in this new version – the inclusion of AET and human impact is exactly what I was hoping to see, and the fact that it improves the predictive performance is indeed satisfactory.

Concerning some of my other comments, I also really appreciate the addition of the species-specific maps, the addition of more validation data, and I now also understand why it has been computationally prohibitive to report full Bayesian uncertainty in the models – I do find the authors' justification to be reasonable.

Concerning the comments of other referees, I also find the authors' responses to these to be reasonable and the solutions well implemented.

Overall, I think that the already attractive manuscript has been further improved, and it is now a solid study that will lead the way to a more integrative cross-scale modelling in ecology, and thus will be cited, widely appreciated, and will look great in Nature Communications.

I only have minor comments.

SPECIFIC COMMENTS

Why are the red-listed species not included in the maps? Isn't it a pity? Anyway, the reason should at least be given, even if briefly.

During the first round of reviews, the request to report a minimum DBH has been made by more than one of us. The authors now give more details on this on lines 485-494, stating that "estimates of species abundance nominally encompass individuals of any size, including seedlings", but I still find this to be vague and confusing. It still does not provide any solid reference for how to interpret the abundance values. How large a seedling needs to be so that it is not a germinating seed any more? And if it is really for all trees, does this match the minimum DBH of the validation datasets?

Lines 64-70: It is unclear in this part if the conditional density concerns each individual tree species, or if it is the density of all tree species together.

Line 160: How about referring to Appendix S4 and to the individual maps somewhere around here?

Line 277: I am not sure what "extinction risk should be estimated rather conservatively" means. Could you try to write it better and more specifically?

Lines 319-322: This is confusing a bit. It is unclear if the entire paper uses these simple models, or if the more complex models will be described later, or in the appendix. You state that "more complex models could be developed", but maybe you should add that the model in your case study is actually such a more complex model. Anyway, this is a really minor comment.

Lines 503-515 in the main text, and Appendix S2. I think that you could give a little bit more detail on the actual model that was used in the main text for the main figures and predictions. I am thinking about including the specific model formula here + some extra descriptions in the main text. Also, maybe this could get its own sub-section. In the current form, it took me some effort to localize the description of the actual model, mostly because it is hidden in the "Model fitting and inference"

section, while maybe it would be better to put the model description in its own sub-section named "Model description", or something like that.

Finally, the code of the model template that is provided (Supplementary Data 1) does not contain the environmental predictors, or does it? Is it the same model that was used to produce the figures and predictions in the main text? I apologize if I am missing something, but maybe it's worth a double check.

Reviewer #3:

Remarks to the Author:

The authors have made many improvements to the manuscript. Thanks. I still think that the paper has much merit, in that it presents a well-thought through analysis of a valuable data set. The implications of the analysis are interesting and relevant to scientific understanding and for management purposes.

However, I fear that the edits have not addressed my principle comments from the first round of review. These concerns are re-raised below, and I have tried to offer suggestions about how the authors could proceed.

Numbering relates to my previous review.

[3-A] Uncertainty: The authors tried to appease the concerns that I, and another reviewer, raised about the lack of uncertainty measures. This was done by suggesting in part of the last paragraph of the discussion that uncertainty is important but is hard. This is a major short-coming of the model and needs to be mentioned and re-iterated throughout the entire manuscript, not just right at the end. It is made worse by the authors use of the model output as though there is no uncertainty. This is dangerous for inference from these data and as a pattern for other researchers to follow. As one example, it is my experience that community metrics, like SADs, are quite variable and that uncertainty will typically imply that one SAD model cannot be distinguished from another.

I understand that there are millions of parameters in the model and that the task is difficult. I do wonder though if there is some precedent in the literature, from other applications of mixed models / empirical Bayes. Such approaches often simplify the task of quantifying uncertainty by using plug-in estimates for the variance components. This is done as the variance components are often estimated using a different likelihood than the random effects (a marginal likelihood or a restricted likelihood) - it is a pragmatic solution. I note that such information is probably already quite readily available from TMB as, I think, the curvature of the joint likelihood at the predicted random effects (of the joint likelihood) will approximate the uncertainty. Isn't this calculated during each and every iteration of the TMB optimisation (to form the Laplace approximation)?

If you are unwilling to pursue this kind of uncertainty quantification, then you could take an intermediate step. Resample the data (with replacement, aka bootstrap) and see how the random effects change. I imagine that you could only do this a few times, due to computation, but it would give some idea. You might even be able to form some other computationally thrifty Monte Carlo measure based on the mode (and Hessian) of parameters.

If you are still unwilling to pursue even this, then please tone down the certainty that each and every inference is made. Along the lines of "subject to unquantified uncertainty".

[3-B] I appreciate that the authors have toned-down the claims that this is a new method. In spite of this, the paper still reads like it is introducing a new method with only an application to floristic data. Evidence of this is in the abstract ("Here, we developed a hierarchical modelling approach to estimate...") and the first line of the discussion ("We developed a methodological framework..."). Also, the abstract and introduction both start with arguments and revision for methods -- the Japanese floristic data doesn't get a mention until well into the second half. When the data, and the ecosystem that it is taken on, are mentioned it is like it exemplifies the method rather than motivates it.

In my previous review, I suggested that the authors either present this as a sensible analysis, albeit undemonstrated about superiority, or present a more formal comparison with other methods. I don't feel that the current manuscript does either of these. Further, the section in the original manuscript that mentioned other methods has been removed in this version.

[3-D] I'm still concerned about model fit. This relates to uncertainty to some extent. The out-of-sample predictions are not strong. This means that it is not obvious that the model is performing well (although we don't know if it still might be performing better than any competitor). I appreciate the inclusion of a qualitative reasoning about the model's comparatively poor performance. However, if the predictive data are qualitatively different, should they be used for validation at all? Would a better strategy be to use cross-validation or a hold-out sample from the original data? I know that this is more computation, but there are some short-cuts to make (like using excellent starting values based on small perturbations from the fit of a previous estimation -- note that it might be that you only need to specify the variance components and fixed effects in starting).

Specific Comments:

[3-New_1] The authors state that the model is computationally demanding. How demanding? How long does it take to run? Is there a way to parallelise some/all of the computation?

[3-New_2] The results of the Poisson-lognormal are not in the Results section. Yet they are referred to in the Discussion. Did I just miss them?

[3-New_3] Do you predict using the occupancy model in addition to the distributional model, or just the distributional model? I can see cases where you would want to do both (e.g. to predict maps, then condition on everything being occupied). However, to predict new data, then the occupancy detection is important. Could this partially explain the observed prediction performance?

The authors use spatially replicated multispecies detection-non-detection data, along with supplementary information on species occurrences in the Japanese archipelago, to infer geographic woody plant species abundances in natural forests at the 10 x 10 km scale of resolution. Multi-species distribution distributions (SADs) for trees are constructed from the data. The constructed SADs are then used to determine the most likely sort of macroevolutionary speciation that has occurred, which showed that protracted speciation (versus point mutation and fission) is the most likely process. The manuscript answers the need, expressed in those papers, for getting beyond local communities to macroscale metacommunity data sets to adequately address the question of what mode of speciation is best, and clearly protracted speciation is able to fit the data very well (Figure 4) and predict reasonable values of metacommunity size, fundamental biodiversity number, speciation rate, and average species lifetimes for different biogeographic regions (Table 2). This seems to be a nice contribution to biodiversity theory.

Comments:

I reviewed an earlier version of this manuscript. The authors have responded to the comments and modified or added text in many places.

A minor question. In Figure 4 the left tail is very short, indicating ‘negligible’ number of rare species (and implying the protracted speciation model). There is evidence from other studies (Magurran and Henderson 2003, Borda-de-Agua et al. 2012) that the number of rare species increases with area, leading to bimodal distribution, in which one of the maxima is for rare species. So there might have been some thought that this trend could continue for larger spatial scales, toward the distribution predicted by the point mutation speciation model. Since this is not observed in the results of this manuscript, is this possibly related to the approach used here, which assumes a homogeneous Poisson distribution?

From a statistical point of view, the “short-tail” result is not an artifact that originated from homogeneous Poisson assumption. (1) Rare species can occur at the regional scale even under the fitted model when their geographic distribution is limited (i.e., when the number of cells with non-zero abundance is small) and/or their abundance is very low. (2) As discussed in the manuscript, the individual density of species will be underestimated when the homogeneity assumption is violated. Given the general tendency of aggregated distribution of plant species, the assumption of a homogeneous Poisson point process is likely to estimate species abundance smaller than it is. Nevertheless, the estimated regional SADs had a short left tail, suggesting that the assumption of a homogeneous Poisson was not the direct cause of the short tail. Furthermore, from a biological point of view, we expect a short left tail in a macroscale SAD because a very small population of a species would be difficult to persist and even unlikely to be documented (this is indeed the fundamental assumption of the protracted speciation model).

Table 1, caption, line 2. Change ‘dataset’ to ‘datasets’

We have revised it as suggested.

Line 64. By ‘conditional individual density’ is ‘conditional on presence’ meant?

We agree with your comment; we have revised this line for clarity, as follows (line 64; please also see the comments of Reviewer 2):

Variation in **the individual density of species, conditional on presence**, and occurrence probability of species were explained by two cell-specific covariates, ...

Lines 67, 68. Change ‘whereas’ to ‘whereas it’

We have revised it as suggested (line 67).

Line 86. By ‘billion’ is the ‘short scale’ 10^9 , or the ‘long scale’ 10^{12} meant?

It is the short scale, 10^9 . We have indicated it in line 86.

Line 429. Change ‘Equations 3 and 9, respectively’ to ‘Equations 9 and 3, respectively’

We have revised it as suggested (line 427).

Line 509. Insert comma after ‘effects’

We have revised it as suggested (line 509).

This work is a strong contribution to the estimation of species abundance distributions. In particular the use of the SADs to generate data on species at the range of low abundances and small areas of occupancy is a useful step towards determining if the species in this range that are not currently classified, might need to be listed if further conditions (e.g., species are declining in abundance or habitat fragmentation is occurring) are met.

We appreciate your constructive comments, which helped us to considerably improve the manuscript.

Reviewer #2 -----

I really liked the first version of the manuscript, and I am impressed by how carefully and rigorously the authors have addressed the extensive comments from the first round of reviews. My main concern was the lack of environmental predictors in the models, and I see this to be fully addressed in this new version – the inclusion of AET and human impact is exactly what I was hoping to see, and the fact that it improves the predictive performance is indeed satisfactory.

Concerning some of my other comments, I also really appreciate the addition of the species-specific maps, the addition of more validation data, and I now also understand why it has been computationally prohibitive to report full Bayesian uncertainty in the models – I do find the authors’ justification to be reasonable.

Concerning the comments of other referees, I also find the authors’ responses to these to be reasonable and the solutions well implemented.

Overall, I think that the already attractive manuscript has been further improved, and it is now a solid study that will lead the way to a more integrative cross-scale modelling in ecology, and thus will be cited, widely appreciated, and will look great in Nature Communications.

I only have minor comments.

We thank you for your previous comments. We believe that the inclusion of the covariates in the model was important. Similarly, the species-specific maps were also highly relevant.

SPECIFIC COMMENTS

Why are the red-listed species not included in the maps? Isn't it a pity? Anyway, the reason should at least be given, even if briefly.

Red-listed species were excluded to eliminate the risk of illegal collection of rare plants. Malicious collectors may use this sort of information, and this is a sensitive issue in Japan. We wanted to avoid an additional pressure on rare species that would possibly occur due to the publication of the abundance maps.

We have mentioned this in Appendix S4 (line 180).

During the first round of reviews, the request to report a minimum DBH has been made by more than one of us. The authors now give more details on this on lines 485-494, stating that “estimates of species abundance nominally encompass individuals of any size, including seedlings”, but I still find this to be vague and confusing. It still does not provide any solid reference for how to interpret the abundance values. How large a seedling needs to be so that it is not a germinating seed any more? And if it is really for all trees, does this match the minimum DBH of the validation datasets?

We have added a brief description that specifies the sizes of the individuals (line 486). A specific criterion discerning germinating seeds and seedlings is foliation of the true leaves. Self-standing individuals with true leaves are both observable and identifiable, thus it is typical in vegetation surveys to count such individuals.

As discussed in the manuscript, the minimum DBH of the validation datasets does not match the nominal definition of individuals in the fitted model. Although this makes validation difficult, in the previous revision, we qualitatively examined how the prediction bias changes as a function of tree size in each validation dataset. A further validation might be possible, at least in principle, using some information on size distribution of individuals in natural forests to adjust model prediction to yield the density of individuals with a specific size range. Given a significant variation in the size distribution of trees (e.g. Wang et al. 2009), detailed data about individual size for each validation plot are required; otherwise, additional uncertainty associated with the adjustment will complicate the interpretation.

Thanks to your comment, we now have a better understanding of the size problem. We have revised the discussion about this topic to improve the statement (lines 205–208). Furthermore, we have added a new figure (Fig. S5) displaying the statistical distribution of individual density in each dataset and model. This is a graphical representation of Table S3 in the previous manuscript, which has been deleted in the current version.

Wang, X., Z. Hao, J. Zhang, J. Lian, B. Li, J. Ye and X. Yao. (2009) Tree size distributions in an old-growth temperate forest. *Oikos* 118:25-36.

Lines 64-70: It is unclear in this part if the conditional density concerns each individual tree species, or if it is the density of all tree species together.

We have revised it as follows (line 64):

Variation in individual density **of species**, ...

Line 160: How about referring to Appendix S4 and to the individual maps somewhere around here?

We decided not to refer to Appendix S4 here. Perhaps, we could refer to Appendix S4 to indicate the geographic distribution of species with such a limited area of occupancy. However, given that we have excluded the map of species with a limited area from Appendix S4, referring to Appendix S4 here will be awkward (we believe, however, that the individual species maps are useful for conservation and should be used in the future in some authorised activities for conservation planning).

Line 277: I am not sure what “extinction risk should be estimated rather conservatively” means. Could you try to write it better and more specifically?

Thank you for your comment. To make this part more concise and clear, we have revised the sentence as follows (lines 276–277):

Given their regional rarity and high endemism, these species may deserve higher conservation priority.

Lines 319-322: This is confusing a bit. I am unclear if the entire paper uses these simple models, or if the more complex models will be described later, or in the appendix. You state that “more complex models could be developed”, but maybe you should add that the model in your case study is actually such a more complex model. Anyway, this is a really minor comment.

We revised this part as follows (lines 319–320):

To describe the basic modelling idea, we here illustrate the models in their simplest form, which includes only intercept and independent random effect terms. The specific and more complex model that we used in our application is described in the following section.

We realise that, as you stated, mentioning possible model extensions here was rather confusing and even unnecessary because they are discussed at the end of the section.

Lines 503-515 in the main text, and Appendix S2. I think that you could give a little bit more detail on the actual model that was used in the main text for the main figures and predictions. I am thinking about including the specific model formula here + some extra descriptions in the main text. Also, maybe this could get its own sub-section. In the current form, it took me some effort to localize the description of the actual model, mostly because it is hidden in the “Model fitting and inference” section, while maybe it would be better to put the model description in its own sub-section named “Model description”, or something like that.

We have described the set of equations that fully specify the fitted model (lines 510–514). Because the model description is rather short and concise, we decided not to provide a specific subsection for it.

Finally, the code of the model template that is provided (Supplementary Data 1) does not contain the environmental predictors, or does it? Is it the same model that was used to produce the figures and predictions in the main text? I apologize if I am missing something, but maybe it’s worth a double check.

Thank you for pointing this. We have replaced the model file.

Reviewer #3 -----

The authors have made many improvements to the manuscript. Thanks. I still think that the paper has much merit, in that it presents a well-thought through analysis of a valuable data set. The implications of the analysis are interesting and relevant to scientific understanding and for management purposes.

However, I fear that the edits have not addressed my principle comments from the first round of review. These concerns are re-raised below, and I have tried to offer suggestions about how the authors could proceed.

Numbering relates to my previous review.

We thank you for providing several constructive comments that have improved our manuscript significantly. To fully incorporate your suggestions, we performed additional analyses, which we have described below.

[3-A] Uncertainty: The authors tried to appease the concerns that I, and another reviewer, raised about the lack of uncertainty measures. This was done by suggesting in part of the last paragraph of the discussion that uncertainty is important but is hard. This is a major short-coming of the model and needs to be mentioned and re-iterated throughout the entire manuscript, not just right at the end. It is made worse by the authors use of the model output as though there is no uncertainty. This is dangerous for inference from these data and as a pattern for other researchers to follow. As one example, it is my experience that community metrics, like SADs, are quite variable and that uncertainty will typically imply that one SAD model cannot be distinguished from another.

I understand that there are millions of parameters in the model and that the task is difficult. I do wonder though if there is some precedent in the literature, from other applications of mixed models / empirical Bayes. Such approaches often simplify the task of quantifying uncertainty by using plug-in estimates for the variance components. This is done as the variance components are often estimated using a different likelihood than the random effects (a marginal likelihood or a restricted likelihood) - it is a pragmatic solution. I note that such information is probably already quite readily available from TMB as, I think, the curvature of the joint likelihood at the predicted random effects (of the joint likelihood) will approximate the uncertainty. Isn't this calculated during each and every iteration of the TMB optimisation (to form the Laplace approximation)?

If you are unwilling to pursue this kind of uncertainty quantification, then you could take an intermediate step. Resample the data (with replacement, aka bootstrap) and see how the random effects change. I imagine that you could only do this a few times, due to computation, but it would give some idea. You might even be able to form some other computationally thrifty Monte Carlo measure based on the mode (and Hessian) of parameters.

If you are still unwilling to pursue even this, then please tone down the certainty that each and every inference is made. Along the lines of "subject to unquantified uncertainty".

Before revising the manuscript, we sent a query. Our question (blue) and the reviewer's response (black) were as follows:

Thank you very much for pointing approaches that can realise reasonable uncertainty assessment. We are going to apply the parametric bootstrapping approach to examine how our inferences could be variable due to sampling error.

This is fine. It is a valid inferential technique. Please take care with the bootstrap of random effects models though -- I think, but do not know, that there are likely to be 'common traps' for bootstrapping those. For what it is worth, I'm not sure that I would carry uncertainty all the way through to the prioritisation step. That seems like a large methodological leap, but I am unsure of the literature. Please make sure that uncertainty is expressed in estimates/maps of richness etc, SADs, and any other similar inferences.

Thank you for the comment. Using the parametric bootstrap procedure for hierarchical models described by Laird and Louis (1987), we evaluated standard errors associated with the MAP estimates of random effects and quantities obtained as a function of these. Accordingly, we revised the manuscript as follows:

- We described the standard errors of the estimates (line 86, Tables 1 and 3).
- We illustrated the standard errors of the estimates (Figures 4 and 5).
- We added a new figure illustrating the standard errors of the estimated community properties (Supplementary Fig. S7).
- We revised the final paragraph of the Discussion (lines 282–304).
- We explained how the uncertainties were assessed using the bootstrap procedure (lines 523–529).

An additional note on the uncertainty estimation is that in the bootstrap procedure, we fixed the auxiliary cell-scale species occurrence information, which directly informs latent variable z (noted in lines 526–529). Although we could have resampled these data like the replicated detection-nondetection data, such a procedure accounts for an extra uncertainty of z for which the specific values are determined based on a given set of auxiliary data. We regard such a procedure rather unreasonable given that under the fitted model, no uncertainty remains in the specific value of z for species and cell with auxiliary occurrence information. We, therefore, evaluated uncertainty in MAP estimates conditional on the given set of auxiliary data. Please note that this sort of additional consideration would not be necessary if we could use the full Bayesian approach in which the posterior distribution of unknown quantities is directly inferred. This is relevant only in the empirical Bayes approach where the uncertainty of MAP estimates is measured in a frequentist manner.

Laird, N. M. and T. A. Louis (1987) Empirical Bayes confidence intervals based on bootstrap samples. *Journal of the American Statistical Association* 82:739-750.

[3-B] I appreciate that the authors have toned-down the claims that this is a new method. In spite of this, the paper still reads like it is introducing a new method with only an application to floristic data. Evidence of this is in the abstract ("Here, we developed a hierarchical modelling approach to estimate...") and the first line of the discussion ("We developed a methodological framework..."). Also, the abstract and introduction both start with arguments and revision for methods -- the Japanese floristic data doesn't get a mention until well into the second half. When the data, and the ecosystem that it is taken on, are mentioned it is like it exemplifies the method rather than motivates it.

In my previous review, I suggested that the authors either present this as a sensible analysis, albeit undemonstrated about superiority, or present a more formal comparison with other methods. I don't

feel that the current manuscript does either of these. Further, the section in the original manuscript that mentioned other methods has been removed in this version.

In the previous round of review, multiple reviewers questioned the novelty of the model we used. We agreed with their opinion and revised the manuscript accordingly, by not emphasising the novelty of the model. In this sense, we believe that the present study is not methodological.

Nevertheless, to the best of our knowledge, such a model has never been used to quantify the abundance of many species at the geographic scale. Illustrating the usability of the model for this kind of inference is, therefore, an important contribution of this study. We call this view as a *modeling approach* (or *framework*), rather than a new methodology, for the estimation of macroscale species abundance, which we have schematically shown in Fig. 1. Accordingly, we intended to describe that the present study illustrates a way to make a relevant ecological inference using an existing methodology, but not propose a new method.

Furthermore, please note that we were not able to compare our approach with other existing methods because we are not aware of similar models that yield estimates of macroscale species abundance.

Yet, in the Discussion, we found three phrases that may give a false impression. We have revised it as follows:

“We developed a methodological framework” to “We developed a modeling approach” (line 165),

“through application of the proposed method to data” to “through the application of this approach with data” (lines 167-168), and

“The method relies on” to “The approach relies on” (line 168).

[3-D] I'm still concerned about model fit. This relates to uncertainty to some extent. The out-of-sample predictions are not strong. This means that it is not obvious that the model is performing well (although we don't know if it still might be performing better than any competitor). I appreciate the inclusion of a qualitative reasoning about the model's comparatively poor performance. However, if the predictive data are qualitatively different, should they be used for validation at all? Would a better strategy be to use cross-validation or a hold-out sample from the original data? I know that this is more computation, but there are some short-cuts to make (like using excellent starting values based on small perturbations from the fit of a previous estimation -- note that it might be that you only need to specify the variance components and fixed effects in starting).

Before revising our manuscript, we had a query. Our question (blue) and the reviewer's response (black) were as follows:

Thank for your suggestion. As an alternative approach, we are wondering if we could obtain the area under the curve (AUC) value to evaluate model fit.

In cross-validation for binary observations (i.e., the occurrence of species in vegetation plots in our case), Bernoulli deviance is typically used as a measure of predictive error. That is, for binary data y and its probability p ,

$$\sum_i (1 - y_i) * \log(1 - p_i) + y_i * \log(p_i).$$

Nevertheless, we are not able to interpret the specific value of the Bernoulli deviance intuitively. Another option for cross-validation may be calculating classification accuracy, but this approach requires an arbitrary threshold probability value to determine species occurrence.

Thus, the cross-validation approach seems to us rather tricky. On the other hand, AUC is widely used to evaluate the fit of binary models, does not require arbitrary thresholds, and intuitive given that it has a maximum value of 1.

For this reason, we are thinking about the use of AUC to evaluate model fit but also wanted to know if this alternative approach fits the original intention of the reviewer.

I can see the reasoning for using AUC. I can also see the reason for using deviance (or log-likelihood). Why not use both? It is a pretty trivial thing to calculate, isn't it?

For what it is worth, I find deviance (or log-likelihood) intuitively easier and probably more informative. I am not alone (Lob et al 2008). It is the relative performance that is important.

Lobo, J. M.; Jiménez-Valverde, A. & Real, R. AUC: a misleading measure of the performance of predictive distribution models *Global ecology and Biogeography*, Blackwell Publishing Ltd, 2008, 17, 145-151

To assess the goodness of model fit, we applied 10-fold cross-validation to the best-fit model in which the AUC values were calculated based on validation (i.e. hold-out) datasets (lines 514–518). As written in the manuscript, we observed a fair cross-validated AUC, which suggests an adequate fit of the model to the vegetation survey dataset. Parallely, we also estimated the Bernoulli deviance and the standardized deviance (Caradima et al. 2019). The former was 459555 (standard deviation: 4254) and the latter was 0.283 (standard deviation: 0.002). Nevertheless, we were not fully confident that these specific values directly inform the goodness of model fit, and hence reported the AUC values but not the deviance and standardized deviance. Although we understand that the AUC is not a versatile measure of model fit, it evaluates a certain aspect of the model fit, namely, the classification performance.

Caradima, B., N. Schuwirth and P. Reichert (2019) From individual to joint species distribution models: A comparison of model complexity and predictive performance. *Journal of Biogeography* 46:2260-2274.

Specific Comments:

[3-New_1] The authors state that the model is computationally demanding. How demanding? How long does it take to run? Is there a way to parallelise some/all of the computation?

The two most computationally demanding steps in our analyses are (1) to fit the hierarchical model to estimate species abundance and (2) to estimate parameters of the protracted speciation model for each ecoregion. We, of course, parallelised these calculations, but the degree of parallelisation has been considerably restricted in step 1 by the amount of computer memory available (even in a supercomputer system) given the large dataset and model. Step 1 takes about 1–3 days and step 2 requires an additional 1–2 days. We can execute about six (but depending on the availability of the system) tasks parallelly, and therefore, we tried the bootstrap and cross-validation procedures described above by fully using our computational resource available for several months.

[3-New_2] The results of the Poisson-lognormal are not in the Results section. Yet they are referred to in the Discussion. Did I just miss them?

The results of the Poisson-lognormal model have been presented only in Table 2, and not in the Results and Discussion because the model is considered as just a statistical (i.e. phenomenological) baseline as stated in "The analyses" subsection under the "Inference of macroevolutionary processes in metacommunities" section. In the revised manuscript, however, we added the Poisson-lognormal results in Fig. 4; we realise that this would be useful for readers to see how the estimated SADs look like (or not like) lognormal.

[3-New_3] Do you predict using the occupancy model in addition to the distributional model, or just the distributional model? I can see cases where you would want to do both (e.g. to predict maps, then condition on everything being occupied). However, to predict new data, then the occupancy detection is important. Could this partially explain the observed prediction performance?

Before revising the manuscript, we had a query. Our question (blue) and the reviewer's response (black) were as follows:

So that we can reply adequately, could you please clarify what the occupancy model and distributional model mean?

Sorry, I can now see that this was confusing. I wanted more clarity around which model you predicted from, and when, and why it was that model. You are proposing a model, which is thinned for detectability with the latent variable z_{ij} . When you predict, do you do so assuming perfect detection ($\psi_{ij}=1$) or imperfect detection ($\psi_{ij}\neq 1$)? The former should be used when making ecological inference, and the latter used when predicting new observations (e.g. cross validation). My previous comment concerned whether these had been mixed up, and prediction was made assuming perfect detection. I doubt it was, but wanted to be sure.

Whilst re-reading sections of your previous manuscript, I carefully read the definition of z_{ij} . I have to wonder what the advantage of defining this random variable at the species level is, as opposed to the individual level? In that case, the Poisson process is thinned (see for example Cressie; 1993), which is itself another Poisson process. This definition could save a substantial amount of computation (due to the thinned result) and it could also be more information rich. I understand that this is a similar approach used to N-mixture models (Royle; 2004).

Cressi, N. Statistics for Spatial Data: Wiley Series in Probability and Statistics Wiley-Interscience New York, 1993

Royle, J. A. N-Mixture Models for Estimating Population Size from Spatially Replicated Counts Biometrics, 2004, 60, 108-115

Thank you for your comment. In the analyses presented in the main text, we used only the best-fit model to make predictions (lines 506–509). Specifically, under the best model, we used equations 12–15 to predict the cell-scale (or regional-scale) community properties. As is clear from the equations, these estimators account for the possibility of $\psi < 1$.

We defined z at the species level, but not at the individual level, simply because it was sufficient for the problem we considered. The individual-level formulation may be an alternative approach to achieve a similar inference, although we are not sure if such a model could provide results much faster than the current model. We consider that the inference of such an individual-based model requires the data augmentation technique, which has been used in capture-recapture studies for abundance estimation to accommodate undetected individuals (Royle et al. 2007, Royle and

Dorazio 2012). This is, in general, a computer-intensive approach, requiring a large number of additional latent variables. We are not confident if this approach is feasible for the problem we considered here, given that it involves billions of individuals.

Royle, J. A., R. M. Dorazio and W. A. Link (2007) Analysis of multinomial models with unknown index using data augmentation. *Journal of Computational and Graphical Statistics* 16:67-85.

Royle, J. A. and R. M. Dorazio (2012) Parameter-expanded data augmentation for Bayesian analysis of capture-recapture models. *Journal of Ornithology* 152:S521-S537.

Additional revisions -----

Besides the revisions described above, we have made some additional minor revisions listed below.

- We have revised Fig. 1 to avoid using an abbreviation for "detection-nondetection (DN)".
- We have revised the axis labels in some figures to indicate logarithms (Fig. 3, 5, S1, S4).
- We have added a reference to a specific subsection (lines 149-150).
- We have added a new reference about the spread of introduced species (Kusumoto et al. 2019) (line 288).
- We have corrected some sentences to improve phrasing (abstract, lines 100-101, 148-149, 166-168).

Reviewers' Comments:

Reviewer #3:

Remarks to the Author:

I wish to thank the authors for engaging with my concerns about their manuscript. I think that the work that it presents is now stronger for this engagement.

I would also like to congratulate the authors for showing the initiative to ask questions, before submitting a revision. This shortened the review process and ensured that everybody was satisfied.

I only have one small comment about the current manuscript: Since the effort has been made to quantify uncertainty, it would be nice to provide some sort of reference to it in the results / discussion. Could the inferences change? Looking at the standard error maps (suppl. figure) and point estimates (figure 2) it seems that there is substantial uncertainty, but not overwhelming as it too-often is in ecology.

Thanks again.

Reviewer #3 -----

I wish to thank the authors for engaging with my concerns about their manuscript. I think that the work that it presents is now stronger for this engagement.

I would also like to congratulate the authors for showing the initiative to ask questions, before submitting a revision. This shortened the review process and ensured that everybody was satisfied.

I only have one small comment about the current manuscript: Since the effort has been made to quantify uncertainty, it would be nice to provide some sort of reference to it in the results / discussion. Could the inferences change? Looking at the standard error maps (suppl. figure) and point estimates (figure 2) it seems that there is substantial uncertainty, but not overwhelming as it too-often is in ecology.

Thanks again.

We appreciate the reviewer's comments and suggestions, including those for our additional queries, that were very helpful to improve the manuscript.

After undertaking careful consideration, we decided not to provide any additional note on the uncertainty in Results or Discussion. We agree with the reviewer that the estimated uncertainty was not that overwhelming. Nevertheless, the standard errors do not formally provide the probability of changing the general conclusion, and therefore, we do not have any solid reasoning to claim if the uncertainty was sufficiently small or not (for example). Standard errors will be adequately used to indicate the uncertainty associated with some specific estimate, but not to infer if the overall inference changes by chance or not. Considering these points, we decided to refer the uncertainty explicitly only where we are reporting the specific value of the estimate (Page 5, line 86).

Thanks again for the constructive discussion.

Additional correction -----

We have made a minor edit on a sentence in Discussion (Page 14, lines 199-201) that could be improved as follows:

[original] Accordingly, the model may effectively predict the density of individuals more than 2 m in height, despite its nominal definition (Supplementary Fig. S5).

[revision] Accordingly, the model may effectively predict, **on average**, the density of individuals **of some specific size**, despite its nominal definition (Supplementary Figure 5).

In addition, we have found an endangered species that should have been removed from the maps in Supplementary Note 4. This species has been included because it has been labeled erroneously as a foreign species. Thus, we have corrected the maps and the descriptions in Supplementary Note 4 appropriately. Figure 5 in the main text has also been updated.